# SHARPNESS-AWARE MINIMIZATION FOR EFFICIENTLY IMPROVING GENERALIZATION

**Pierre Foret** *
Google Research
pierreforet@google.com

**Ariel Kleiner**
Google Research
akleiner@google.com

**Hossein Mobahi**
Google Research
hmobahi@google.com

**Behnam Neyshabur**
Blueshift, Alphabet
neyshabur@google.com

## ABSTRACT

In today's heavily overparameterized models, the value of the training loss provides few guarantees on model generalization ability. Indeed, optimizing only the training loss value, as is commonly done, can easily lead to suboptimal model quality. Motivated by prior work connecting the geometry of the loss landscape and generalization, we introduce a novel, effective procedure for instead simultaneously minimizing loss value and loss sharpness. In particular, our procedure, Sharpness-Aware Minimization (SAM), seeks parameters that lie in neighborhoods having uniformly low loss; this formulation results in a min-max optimization problem on which gradient descent can be performed efficiently. We present empirical results showing that SAM improves model generalization across a variety of benchmark datasets (e.g., CIFAR-{10, 100}, ImageNet, finetuning tasks) and models, yielding novel state-of-the-art performance for several. Additionally, we find that SAM natively provides robustness to label noise on par with that provided by state-of-the-art procedures that specifically target learning with noisy labels. We open source our code at https://github.com/google-research/sam.

## 1 INTRODUCTION

Modern machine learning's success in achieving ever better performance on a wide range of tasks has relied in significant part on ever heavier overparameterization, in conjunction with developing ever more effective training algorithms that are able to find parameters that generalize well. Indeed, many modern neural networks can easily memorize the training data and have the capacity to readily overfit (Zhang et al., 2016). Such heavy overparameterization is currently required to achieve state-of-the-art results in a variety of domains (Tan & Le, 2019; Kolesnikov et al., 2020; Huang et al., 2018). In turn, it is essential that such models be trained using procedures that ensure that the parameters actually selected do in fact generalize beyond the training set.

Unfortunately, simply minimizing commonly used loss functions (e.g., cross-entropy) on the training set is typically not sufficient to achieve satisfactory generalization. The training loss landscapes of today's models are commonly complex and non-convex, with a multiplicity of local and global minima, and with different global minima yielding models with different generalization abilities (Shirish Keskar et al., 2016). As a result, the choice of optimizer (and associated optimizer settings) from among the many available (e.g., stochastic gradient descent (Nesterov, 1983), Adam (Kingma & Ba, 2014), RMSProp (Hinton et al.), and others (Duchi et al., 2011; Dozat, 2016; Martens & Grosse, 2015)) has become an important design choice, though understanding of its relationship to model generalization remains nascent (Shirish Keskar et al., 2016; Wilson et al., 2017; Shirish Keskar & Socher, 2017; Agarwal et al., 2020; Jacot et al., 2018). Relatedly, a panoply of methods for modifying the training process have been proposed, including dropout (Srivastava et al., 2014),

---

*Work done as part of the Google AI Residency program.

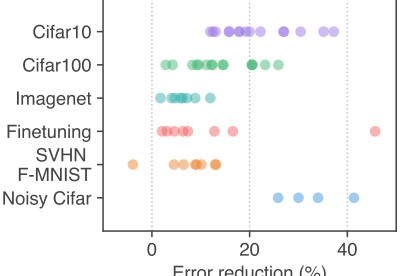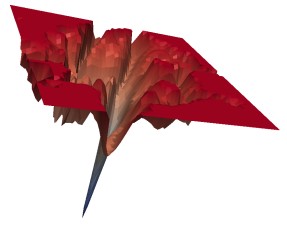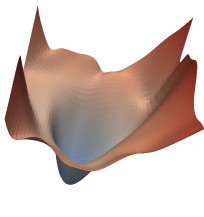

Figure 1: (left) Error rate reduction obtained by switching to SAM. Each point is a different dataset / model / data augmentation. (middle) A sharp minimum to which a ResNet trained with SGD converged. (right) A wide minimum to which the same ResNet trained with SAM converged.

batch normalization (Ioffe & Szegedy, 2015), stochastic depth (Huang et al., 2016), data augmentation (Cubuk et al., 2018), and mixed sample augmentations (Zhang et al., 2017; Harris et al., 2020).

The connection between the geometry of the loss landscape—in particular, the flatness of minima—and generalization has been studied extensively from both theoretical and empirical perspectives (Shirish Keskar et al., 2016; Dziugaite & Roy, 2017; Jiang et al., 2019). While this connection has held the promise of enabling new approaches to model training that yield better generalization, practical efficient algorithms that specifically seek out flatter minima and furthermore effectively improve generalization on a range of state-of-the-art models have thus far been elusive (e.g., see (Chaudhari et al., 2016; Izmailov et al., 2018); we include a more detailed discussion of prior work in Section 5).

We present here a new efficient, scalable, and effective approach to improving model generalization ability that directly leverages the geometry of the loss landscape and its connection to generalization, and is powerfully complementary to existing techniques. In particular, we make the following contributions:

- We introduce Sharpness-Aware Minimization (SAM), a novel procedure that improves model generalization by simultaneously minimizing loss value and loss sharpness. SAM functions by seeking parameters that lie in neighborhoods having uniformly low loss value (rather than parameters that only themselves have low loss value, as illustrated in the middle and righthand images of Figure 1), and can be implemented efficiently and easily.

- We show via a rigorous empirical study that using SAM improves model generalization ability across a range of widely studied computer vision tasks (e.g., CIFAR-{10, 100}, ImageNet, finetuning tasks) and models, as summarized in the lefthand plot of Figure 1. For example, applying SAM yields novel state-of-the-art performance for a number of already-intensely-studied tasks, such as ImageNet, CIFAR-{10, 100}, SVHN, Fashion-MNIST, and the standard set of image classification finetuning tasks (e.g., Flowers, Stanford Cars, Oxford Pets, etc).

- We show that SAM furthermore provides robustness to label noise on par with that provided by state-of-the-art procedures that specifically target learning with noisy labels.

- Through the lens provided by SAM, we further elucidate the connection between loss sharpness and generalization by surfacing a promising new notion of sharpness, which we term *m-sharpness*.

Section 2 below derives the SAM procedure and presents the resulting algorithm in full detail. Section 3 evaluates SAM empirically, and Section 4 further analyzes the connection between loss sharpness and generalization through the lens of SAM. Finally, we conclude with an overview of related work and a discussion of conclusions and future work in Sections 5 and 6, respectively.

## 2 SHARPNESS-AWARE MINIMIZATION (SAM)

Throughout the paper, we denote scalars as $a$, vectors as $\boldsymbol{a}$, matrices as $\boldsymbol{A}$, sets as $\mathcal{A}$, and equality by definition as $\triangleq$. Given a training dataset $\mathcal{S} \triangleq \cup_{i=1}^{n}\{(\boldsymbol{x}_i, \boldsymbol{y}_i)\}$ drawn i.i.d. from distribution $\mathscr{D}$, we seek to learn a model that generalizes well. In particular, consider a family of models parameterized by $\boldsymbol{w} \in \mathcal{W} \subseteq \mathbb{R}^d$; given a per-data-point loss function $l : \mathcal{W} \times \mathcal{X} \times \mathcal{Y} \rightarrow \mathbb{R}_+$, we define the training set loss $L_S(\boldsymbol{w}) \triangleq \frac{1}{n}\sum_{i=1}^{n} l(\boldsymbol{w}, \boldsymbol{x}_i, \boldsymbol{y}_i)$ and the population loss $L_{\mathscr{D}}(\boldsymbol{w}) \triangleq \mathbb{E}_{(\boldsymbol{x}, \boldsymbol{y}) \sim D}[l(\boldsymbol{w}, \boldsymbol{x}, \boldsymbol{y})]$. Having observed only $\mathcal{S}$, the goal of model training is to select model parameters $\boldsymbol{w}$ having low population loss $L_{\mathscr{D}}(\boldsymbol{w})$.

Utilizing $L_{\mathcal{S}}(\boldsymbol{w})$ as an estimate of $L_{\mathscr{D}}(\boldsymbol{w})$ motivates the standard approach of selecting parameters $\boldsymbol{w}$ by solving $\min_{\boldsymbol{w}} L_{\mathcal{S}}(\boldsymbol{w})$ (possibly in conjunction with a regularizer on $\boldsymbol{w}$) using an optimization procedure such as SGD or Adam. Unfortunately, however, for modern overparameterized models such as deep neural networks, typical optimization approaches can easily result in suboptimal performance at test time. In particular, for modern models, $L_{\mathcal{S}}(\boldsymbol{w})$ is typically non-convex in $\boldsymbol{w}$, with multiple local and even global minima that may yield similar values of $L_{\mathcal{S}}(\boldsymbol{w})$ while having significantly different generalization performance (i.e., significantly different values of $L_{\mathscr{D}}(\boldsymbol{w})$).

Motivated by the connection between sharpness of the loss landscape and generalization, we propose a different approach: rather than seeking out parameter values $\boldsymbol{w}$ that simply have low training loss value $L_{\mathcal{S}}(\boldsymbol{w})$, we seek out parameter values whose entire neighborhoods have uniformly low training loss value (equivalently, neighborhoods having both low loss and low curvature). The following theorem illustrates the motivation for this approach by bounding generalization ability in terms of neighborhood-wise training loss (full theorem statement and proof in Appendix A):

**Theorem (stated informally) 1.** *For any $\rho > 0$, with high probability over training set $\mathcal{S}$ generated from distribution $\mathscr{D}$,*

$$L_{\mathscr{D}}(\boldsymbol{w}) \leq \max_{\|\boldsymbol{\epsilon}\|_2 \leq \rho} L_{\mathcal{S}}(\boldsymbol{w} + \boldsymbol{\epsilon}) + h(\|\boldsymbol{w}\|_2^2/\rho^2),$$

*where $h : \mathbb{R}_+ \rightarrow \mathbb{R}_+$ is a strictly increasing function (under some technical conditions on $L_{\mathscr{D}}(\boldsymbol{w})$).*

To make explicit our sharpness term, we can rewrite the right hand side of the inequality above as

$$\left[\max_{\|\boldsymbol{\epsilon}\|_2 \leq \rho} L_{\mathcal{S}}(\boldsymbol{w} + \boldsymbol{\epsilon}) - L_{\mathcal{S}}(\boldsymbol{w})\right] + L_{\mathcal{S}}(\boldsymbol{w}) + h(\|\boldsymbol{w}\|_2^2/\rho^2).$$

The term in square brackets captures the sharpness of $L_{\mathcal{S}}$ at $\boldsymbol{w}$ by measuring how quickly the training loss can be increased by moving from $\boldsymbol{w}$ to a nearby parameter value; this sharpness term is then summed with the training loss value itself and a regularizer on the magnitude of $\boldsymbol{w}$. Given that the specific function $h$ is heavily influenced by the details of the proof, we substitute the second term with $\lambda\|w\|_2^2$ for a hyperparameter $\lambda$, yielding a standard L2 regularization term. Thus, inspired by the terms from the bound, we propose to select parameter values by solving the following Sharpness-Aware Minimization (SAM) problem:

$$\min_{\boldsymbol{w}} L_{\mathcal{S}}^{SAM}(\boldsymbol{w}) + \lambda\|\boldsymbol{w}\|_2^2 \quad \text{where} \quad L_{\mathcal{S}}^{SAM}(\boldsymbol{w}) \triangleq \max_{\|\boldsymbol{\epsilon}\|_p \leq \rho} L_{\mathcal{S}}(\boldsymbol{w} + \boldsymbol{\epsilon}), \tag{1}$$

where $\rho \geq 0$ is a hyperparameter and $p \in [1, \infty]$ (we have generalized slightly from an L2-norm to a $p$-norm in the maximization over $\boldsymbol{\epsilon}$, though we show empirically in appendix C.5 that $p = 2$ is typically optimal). Figure 1 shows[1] the loss landscape for a model that converged to minima found by minimizing either $L_{\mathcal{S}}(\boldsymbol{w})$ or $L_{\mathcal{S}}^{SAM}(\boldsymbol{w})$, illustrating that the sharpness-aware loss prevents the model from converging to a sharp minimum.

In order to minimize $L_{\mathcal{S}}^{SAM}(\boldsymbol{w})$, we derive an efficient and effective approximation to $\nabla_{\boldsymbol{w}} L_{\mathcal{S}}^{SAM}(\boldsymbol{w})$ by differentiating through the inner maximization, which in turn enables us to apply stochastic gradient descent directly to the SAM objective. Proceeding down this path, we first approximate the inner maximization problem via a first-order Taylor expansion of $L_{\mathcal{S}}(\boldsymbol{w} + \boldsymbol{\epsilon})$ w.r.t. $\boldsymbol{\epsilon}$ around $\boldsymbol{0}$, obtaining

$$\boldsymbol{\epsilon}^*(\boldsymbol{w}) \triangleq \underset{\|\boldsymbol{\epsilon}\|_p \leq \rho}{\arg\max}\, L_{\mathcal{S}}(\boldsymbol{w} + \boldsymbol{\epsilon}) \approx \underset{\|\boldsymbol{\epsilon}\|_p \leq \rho}{\arg\max}\, L_{\mathcal{S}}(\boldsymbol{w}) + \boldsymbol{\epsilon}^T \nabla_{\boldsymbol{w}} L_{\mathcal{S}}(\boldsymbol{w}) = \underset{\|\boldsymbol{\epsilon}\|_p \leq \rho}{\arg\max}\, \boldsymbol{\epsilon}^T \nabla_{\boldsymbol{w}} L_{\mathcal{S}}(\boldsymbol{w}).$$

---

[1]Figure 1 was generated following Li et al. (2017) with the provided ResNet56 (no residual connections) checkpoint, and training the same model with SAM.

In turn, the value $\hat{\epsilon}(\boldsymbol{w})$ that solves this approximation is given by the solution to a classical dual norm problem ($|\cdot|^{q-1}$ denotes elementwise absolute value and power)[2]:

$$\hat{\epsilon}(\boldsymbol{w}) = \rho \, \text{sign} \left(\nabla_{\boldsymbol{w}} L_{\mathcal{S}}(\boldsymbol{w})\right) |\nabla_{\boldsymbol{w}} L_{\mathcal{S}}(\boldsymbol{w})|^{q-1} \Big/ \left(\|\nabla_{\boldsymbol{w}} L_{\mathcal{S}}(\boldsymbol{w})\|_q^q\right)^{1/p} \tag{2}$$

where $1/p + 1/q = 1$. Substituting back into equation (1) and differentiating, we then have

$$\nabla_{\boldsymbol{w}} L_{\mathcal{S}}^{SAM}(\boldsymbol{w}) \approx \nabla_{\boldsymbol{w}} L_{\mathcal{S}}(\boldsymbol{w} + \hat{\epsilon}(\boldsymbol{w})) = \frac{d(\boldsymbol{w} + \hat{\epsilon}(\boldsymbol{w}))}{d\boldsymbol{w}} \nabla_{\boldsymbol{w}} L_{\mathcal{S}}(\boldsymbol{w})|_{\boldsymbol{w} + \hat{\epsilon}(w)}$$

$$= \nabla_w L_{\mathcal{S}}(\boldsymbol{w})|_{\boldsymbol{w} + \hat{\epsilon}(\boldsymbol{w})} + \frac{d\hat{\epsilon}(\boldsymbol{w})}{d\boldsymbol{w}} \nabla_{\boldsymbol{w}} L_{\mathcal{S}}(\boldsymbol{w})|_{\boldsymbol{w} + \hat{\epsilon}(\boldsymbol{w})}.$$

This approximation to $\nabla_{\boldsymbol{w}} L_{\mathcal{S}}^{SAM}(\boldsymbol{w})$ can be straightforwardly computed via automatic differentiation, as implemented in common libraries such as JAX, TensorFlow, and PyTorch. Though this computation implicitly depends on the Hessian of $L_{\mathcal{S}}(\boldsymbol{w})$ because $\hat{\epsilon}(\boldsymbol{w})$ is itself a function of $\nabla_{\boldsymbol{w}} L_{\mathcal{S}}(\boldsymbol{w})$, the Hessian enters only via Hessian-vector products, which can be computed tractably without materializing the Hessian matrix. Nonetheless, to further accelerate the computation, we drop the second-order terms. obtaining our final gradient approximation:

$$\nabla_{\boldsymbol{w}} L_{\mathcal{S}}^{SAM}(\boldsymbol{w}) \approx \nabla_{\boldsymbol{w}} L_{\mathcal{S}}(w)|_{\boldsymbol{w} + \hat{\epsilon}(\boldsymbol{w})}. \tag{3}$$

As shown by the results in Section 3, this approximation (without the second-order terms) yields an effective algorithm. In Appendix C.4, we additionally investigate the effect of instead including the second-order terms; in that initial experiment, including them surprisingly degrades performance, and further investigating these terms' effect should be a priority in future work.

We obtain the final SAM algorithm by applying a standard numerical optimizer such as stochastic gradient descent (SGD) to the SAM objective $L_{\mathcal{S}}^{SAM}(\boldsymbol{w})$, using equation 3 to compute the requisite objective function gradients. Algorithm 1 gives pseudo-code for the full SAM algorithm, using SGD as the base optimizer, and Figure 2 schematically illustrates a single SAM parameter update.

**Input:** Training set $\mathcal{S} \triangleq \cup_{i=1}^n \{(\boldsymbol{x}_i, \boldsymbol{y}_i)\}$, Loss function
$\quad\quad l : \mathcal{W} \times \mathcal{X} \times \mathcal{Y} \to \mathbb{R}_+$, Batch size $b$, Step size $\eta > 0$,
$\quad\quad$ Neighborhood size $\rho > 0$.
**Output:** Model trained with SAM
Initialize weights $\boldsymbol{w}_0$, $t = 0$;
**while** *not converged* **do**
$\quad$ Sample batch $\mathcal{B} = \{(\boldsymbol{x}_1, \boldsymbol{y}_1), ...(\boldsymbol{x}_b, \boldsymbol{y}_b)\}$;
$\quad$ Compute gradient $\nabla_{\boldsymbol{w}} L_{\mathcal{B}}(\boldsymbol{w})$ of the batch's training loss;
$\quad$ Compute $\hat{\epsilon}(\boldsymbol{w})$ per equation 2;
$\quad$ Compute gradient approximation for the SAM objective
$\quad\quad$ (equation 3): $\boldsymbol{g} = \nabla_w L_{\mathcal{B}}(\boldsymbol{w})|_{\boldsymbol{w} + \hat{\epsilon}(\boldsymbol{w})}$;
$\quad$ Update weights: $\boldsymbol{w}_{t+1} = \boldsymbol{w}_t - \eta \boldsymbol{g}$;
$\quad$ $t = t + 1$;
**end**
**return** $\boldsymbol{w}_t$

**Algorithm 1:** SAM algorithm

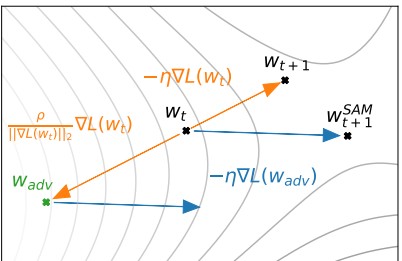

Figure 2: Schematic of the SAM parameter update.

## 3 EMPIRICAL EVALUATION

In order to assess SAM's efficacy, we apply it to a range of different tasks, including image classification from scratch (including on CIFAR-10, CIFAR-100, and ImageNet), finetuning pretrained models, and learning with noisy labels. In all cases, we measure the benefit of using SAM by simply replacing the optimization procedure used to train existing models with SAM, and computing the resulting effect on model generalization. As seen below, SAM materially improves generalization performance in the vast majority of these cases.

---

[2]In the case of interest $p = 2$, this boils down to simply rescaling the gradient such that its norm is $\rho$.

## 3.1 IMAGE CLASSIFICATION FROM SCRATCH

We first evaluate SAM's impact on generalization for today's state-of-the-art models on CIFAR-10 and CIFAR-100 (without pretraining): WideResNets with ShakeShake regularization (Zagoruyko & Komodakis, 2016; Gastaldi, 2017) and PyramidNet with ShakeDrop regularization (Han et al., 2016; Yamada et al., 2018). Note that some of these models have already been heavily tuned in prior work and include carefully chosen regularization schemes to prevent overfitting; therefore, significantly improving their generalization is quite non-trivial. We have ensured that our implementations' generalization performance in the absence of SAM matches or exceeds that reported in prior work (Cubuk et al., 2018; Lim et al., 2019)

All results use basic data augmentations (horizontal flip, padding by four pixels, and random crop). We also evaluate in the setting of more advanced data augmentation methods such as cutout regularization (Devries & Taylor, 2017) and AutoAugment (Cubuk et al., 2018), which are utilized by prior work to achieve state-of-the-art results.

SAM has a single hyperparameter $\rho$ (the neighborhood size), which we tune via a grid search over $\{0.01, 0.02, 0.05, 0.1, 0.2, 0.5\}$ using 10% of the training set as a validation set[3]. Please see appendix C.1 for the values of all hyperparameters and additional training details. As each SAM weight update requires two backpropagation operations (one to compute $\hat{\epsilon}(w)$ and another to compute the final gradient), we allow each non-SAM training run to execute twice as many epochs as each SAM training run, and we report the best score achieved by each non-SAM training run across either the standard epoch count or the doubled epoch count[4]. We run five independent replicas of each experimental condition for which we report results (each with independent weight initialization and data shuffling), reporting the resulting mean error (or accuracy) on the test set, and the associated 95% confidence interval. Our implementations utilize JAX (Bradbury et al., 2018), and we train all models on a single host having 8 Nvidia V100 GPUs[5]. To compute the SAM update when parallelizing across multiple accelerators, we divide each data batch evenly among the accelerators, independently compute the SAM gradient on each accelerator, and average the resulting sub-batch SAM gradients to obtain the final SAM update.

As seen in Table 1, SAM improves generalization across all settings evaluated for CIFAR-10 and CIFAR-100. For example, SAM enables a simple WideResNet to attain 1.6% test error, versus 2.2% error without SAM. Such gains have previously been attainable only by using more complex model architectures (e.g., PyramidNet) and regularization schemes (e.g., Shake-Shake, ShakeDrop); SAM provides an easily-implemented, model-independent alternative. Furthermore, SAM delivers improvements even when applied atop complex architectures that already use sophisticated regularization: for instance, applying SAM to a PyramidNet with ShakeDrop regularization yields 10.3% error on CIFAR-100, which is, to our knowledge, a new state-of-the-art on this dataset without the use of additional data.

Beyond CIFAR-{10, 100}, we have also evaluated SAM on the SVHN (Netzer et al., 2011) and Fashion-MNIST datasets (Xiao et al., 2017). Once again, SAM enables a simple WideResNet to achieve accuracy at or above the state-of-the-art for these datasets: 0.99% error for SVHN, and 3.59% for Fashion-MNIST. Details are available in appendix B.1.

To assess SAM's performance at larger scale, we apply it to ResNets (He et al., 2015) of different depths (50, 101, 152) trained on ImageNet (Deng et al., 2009). In this setting, following prior work (He et al., 2015; Szegedy et al., 2015), we resize and crop images to 224-pixel resolution, normalize them, and use batch size 4096, initial learning rate 1.0, cosine learning rate schedule, SGD optimizer with momentum 0.9, label smoothing of 0.1, and weight decay 0.0001. When applying SAM, we use $\rho = 0.05$ (determined via a grid search on ResNet-50 trained for 100 epochs). We train all models on ImageNet for up to 400 epochs using a Google Cloud TPUv3 and report top-1 and top-5 test error rates for each experimental condition (mean and 95% confidence interval across 5 independent runs).

---

[3]We found $\rho = 0.05$ to be a solid default value, and we report in appendix C.3 the scores for all our experiments, obtained with $\rho = 0.05$ without further tuning.

[4]Training for longer generally did not improve accuracy significantly, except for the models previously trained for only 200 epochs and for the largest, most regularized model (PyramidNet + ShakeDrop).

[5]Because SAM's performance is amplified by not syncing the perturbations, data parallelism is highly recommended to leverage SAM's full potential (see Section 4 for more details).

| | | CIFAR-10 | | CIFAR-100 | |
|---|---|---|---|---|---|
| Model | Augmentation | SAM | SGD | SAM | SGD |
| WRN-28-10 (200 epochs) | Basic | $\mathbf{2.7}_{\pm 0.1}$ | $3.5_{\pm 0.1}$ | $\mathbf{16.5}_{\pm 0.2}$ | $18.8_{\pm 0.2}$ |
| WRN-28-10 (200 epochs) | Cutout | $\mathbf{2.3}_{\pm 0.1}$ | $2.6_{\pm 0.1}$ | $\mathbf{14.9}_{\pm 0.2}$ | $16.9_{\pm 0.1}$ |
| WRN-28-10 (200 epochs) | AA | $\mathbf{2.1}_{\pm <0.1}$ | $2.3_{\pm 0.1}$ | $\mathbf{13.6}_{\pm 0.2}$ | $15.8_{\pm 0.2}$ |
| WRN-28-10 (1800 epochs) | Basic | $\mathbf{2.4}_{\pm 0.1}$ | $3.5_{\pm 0.1}$ | $\mathbf{16.3}_{\pm 0.2}$ | $19.1_{\pm 0.1}$ |
| WRN-28-10 (1800 epochs) | Cutout | $\mathbf{2.1}_{\pm 0.1}$ | $2.7_{\pm 0.1}$ | $\mathbf{14.0}_{\pm 0.1}$ | $17.4_{\pm 0.1}$ |
| WRN-28-10 (1800 epochs) | AA | $\mathbf{1.6}_{\pm 0.1}$ | $2.2_{\pm <0.1}$ | $\mathbf{12.8}_{\pm 0.2}$ | $16.1_{\pm 0.2}$ |
| Shake-Shake (26 2x96d) | Basic | $\mathbf{2.3}_{\pm <0.1}$ | $2.7_{\pm 0.1}$ | $\mathbf{15.1}_{\pm 0.1}$ | $17.0_{\pm 0.1}$ |
| Shake-Shake (26 2x96d) | Cutout | $\mathbf{2.0}_{\pm <0.1}$ | $2.3_{\pm 0.1}$ | $\mathbf{14.2}_{\pm 0.2}$ | $15.7_{\pm 0.2}$ |
| Shake-Shake (26 2x96d) | AA | $\mathbf{1.6}_{\pm <0.1}$ | $1.9_{\pm 0.1}$ | $\mathbf{12.8}_{\pm 0.1}$ | $14.1_{\pm 0.2}$ |
| PyramidNet | Basic | $\mathbf{2.7}_{\pm 0.1}$ | $4.0_{\pm 0.1}$ | $\mathbf{14.6}_{\pm 0.4}$ | $19.7_{\pm 0.3}$ |
| PyramidNet | Cutout | $\mathbf{1.9}_{\pm 0.1}$ | $2.5_{\pm 0.1}$ | $\mathbf{12.6}_{\pm 0.2}$ | $16.4_{\pm 0.1}$ |
| PyramidNet | AA | $\mathbf{1.6}_{\pm 0.1}$ | $1.9_{\pm 0.1}$ | $\mathbf{11.6}_{\pm 0.1}$ | $14.6_{\pm 0.1}$ |
| PyramidNet+ShakeDrop | Basic | $\mathbf{2.1}_{\pm 0.1}$ | $2.5_{\pm 0.1}$ | $\mathbf{13.3}_{\pm 0.2}$ | $14.5_{\pm 0.1}$ |
| PyramidNet+ShakeDrop | Cutout | $\mathbf{1.6}_{\pm <0.1}$ | $1.9_{\pm 0.1}$ | $\mathbf{11.3}_{\pm 0.1}$ | $11.8_{\pm 0.2}$ |
| PyramidNet+ShakeDrop | AA | $\mathbf{1.4}_{\pm <0.1}$ | $1.6_{\pm <0.1}$ | $\mathbf{10.3}_{\pm 0.1}$ | $10.6_{\pm 0.1}$ |

Table 1: Results for SAM on state-of-the-art models on CIFAR-{10, 100} (WRN = WideResNet; AA = AutoAugment; SGD is the standard non-SAM procedure used to train these models).

As seen in Table 2, SAM again consistently improves performance, for example improving the ImageNet top-1 error rate of ResNet-152 from 20.3% to 18.4%. Furthermore, note that SAM enables increasing the number of training epochs while continuing to improve accuracy without overfitting. In contrast, the standard training procedure (without SAM) generally significantly overfits as training extends from 200 to 400 epochs.

| Model | Epoch | SAM | | Standard Training (No SAM) | |
|---|---|---|---|---|---|
| | | Top-1 | Top-5 | Top-1 | Top-5 |
| ResNet-50 | 100 | $\mathbf{22.5}_{\pm 0.1}$ | $6.28_{\pm 0.08}$ | $22.9_{\pm 0.1}$ | $6.62_{\pm 0.11}$ |
| | 200 | $\mathbf{21.4}_{\pm 0.1}$ | $5.82_{\pm 0.03}$ | $22.3_{\pm 0.1}$ | $6.37_{\pm 0.04}$ |
| | 400 | $\mathbf{20.9}_{\pm 0.1}$ | $5.51_{\pm 0.03}$ | $22.3_{\pm 0.1}$ | $6.40_{\pm 0.06}$ |
| ResNet-101 | 100 | $\mathbf{20.2}_{\pm 0.1}$ | $5.12_{\pm 0.03}$ | $21.2_{\pm 0.1}$ | $5.66_{\pm 0.05}$ |
| | 200 | $\mathbf{19.4}_{\pm 0.1}$ | $4.76_{\pm 0.03}$ | $20.9_{\pm 0.1}$ | $5.66_{\pm 0.04}$ |
| | 400 | $\mathbf{19.0}_{\pm <0.01}$ | $4.65_{\pm 0.05}$ | $22.3_{\pm 0.1}$ | $6.41_{\pm 0.06}$ |
| ResNet-152 | 100 | $\mathbf{19.2}_{\pm <0.01}$ | $4.69_{\pm 0.04}$ | $20.4_{\pm <0.0}$ | $5.39_{\pm 0.06}$ |
| | 200 | $\mathbf{18.5}_{\pm 0.1}$ | $4.37_{\pm 0.03}$ | $20.3_{\pm 0.2}$ | $5.39_{\pm 0.07}$ |
| | 400 | $\mathbf{18.4}_{\pm <0.01}$ | $4.35_{\pm 0.04}$ | $20.9_{\pm <0.0}$ | $5.84_{\pm 0.07}$ |

Table 2: Test error rates for ResNets trained on ImageNet, with and without SAM.

## 3.2 FINETUNING

Transfer learning by pretraining a model on a large related dataset and then finetuning on a smaller target dataset of interest has emerged as a powerful and widely used technique for producing high-quality models for a variety of different tasks. We show here that SAM once again offers considerable benefits in this setting, even when finetuning extremely large, state-of-the-art, already high-performing models.

In particular, we apply SAM to finetuning EfficentNet-b7 (pretrained on ImageNet) and EfficientNet-L2 (pretrained on ImageNet plus unlabeled JFT; input resolution 475) (Tan & Le, 2019; Kornblith et al., 2018; Huang et al., 2018). We initialize these models to publicly available checkpoints[6] trained with RandAugment (84.7% accuracy on ImageNet) and NoisyStudent (88.2% accuracy on ImageNet), respectively. We finetune these models on each of several target datasets by training each model starting from the aforementioned checkpoint; please see the appendix for details of the hyperparameters used. We report the mean and 95% confidence interval of top-1 test error over 5 independent runs for each dataset.

---

[6] https://github.com/tensorflow/tpu/tree/master/models/official/efficientnet

As seen in Table 3, SAM uniformly improves performance relative to finetuning without SAM. Furthermore, in many cases, SAM yields novel state-of-the-art performance, including 0.30% error on CIFAR-10, 3.92% error on CIFAR-100, and 11.39% error on ImageNet.

| Dataset | EffNet-b7 + SAM | EffNet-b7 | Prev. SOTA (ImageNet only) | EffNet-L2 + SAM | EffNet-L2 | Prev. SOTA |
|---|---|---|---|---|---|---|
| FGVC_Aircraft | $6.80_{\pm0.06}$ | $8.15_{\pm0.08}$ | **5.3** (TBMSL-Net) | $4.82_{\pm0.08}$ | $5.80_{\pm0.1}$ | 5.3 (TBMSL-Net) |
| Flowers | $0.63_{\pm0.02}$ | $1.16_{\pm0.05}$ | 0.7 (BiT-M) | $0.35_{\pm0.01}$ | $0.40_{\pm0.02}$ | 0.37 (EffNet) |
| Oxford_IIIT_Pets | $3.97_{\pm0.04}$ | $4.24_{\pm0.09}$ | 4.1 (Gpipe) | $2.90_{\pm0.04}$ | $3.08_{\pm0.04}$ | 4.1 (Gpipe) |
| Stanford_Cars | $5.18_{\pm0.02}$ | $5.94_{\pm0.06}$ | **5.0** (TBMSL-Net) | $4.04_{\pm0.03}$ | $4.93_{\pm0.04}$ | **3.8** (DAT) |
| CIFAR-10 | $0.88_{\pm0.02}$ | $0.95_{\pm0.03}$ | 1 (Gpipe) | $0.30_{\pm0.01}$ | $0.34_{\pm0.02}$ | 0.63 (BiT-L) |
| CIFAR-100 | $7.44_{\pm0.06}$ | $7.68_{\pm0.06}$ | 7.83 (BiT-M) | $3.92_{\pm0.06}$ | $4.07_{\pm0.08}$ | 6.49 (BiT-L) |
| Birdsnap | $13.64_{\pm0.15}$ | $14.30_{\pm0.18}$ | 15.7 (EffNet) | $9.93_{\pm0.15}$ | $10.31_{\pm0.15}$ | 14.5 (DAT) |
| Food101 | $7.02_{\pm0.02}$ | $7.17_{\pm0.03}$ | 7.0 (Gpipe) | $3.82_{\pm0.01}$ | $3.97_{\pm0.03}$ | 4.7 (DAT) |
| ImageNet | $15.14_{\pm0.03}$ | 15.3 | **14.2** (KDforAA) | $11.39_{\pm0.02}$ | 11.8 | 11.45 (ViT) |

Table 3: Top-1 error rates for finetuning EfficientNet-b7 (left; ImageNet pretraining only) and EfficientNet-L2 (right; pretraining on ImageNet plus additional data, such as JFT) on various downstream tasks. Previous state-of-the-art (SOTA) includes EfficientNet (EffNet) (Tan & Le, 2019), Gpipe (Huang et al., 2018), DAT (Ngiam et al., 2018), BiT-M/L (Kolesnikov et al., 2020), KD-forAA (Wei et al., 2020), TBMSL-Net (Zhang et al., 2020), and ViT (Dosovitskiy et al., 2020).

## 3.3 ROBUSTNESS TO LABEL NOISE

The fact that SAM seeks out model parameters that are robust to perturbations suggests SAM's potential to provide robustness to noise in the training set (which would perturb the training loss landscape). Thus, we assess here the degree of robustness that SAM provides to label noise.

In particular, we measure the effect of applying SAM in the classical noisy-label setting for CIFAR-10, in which a fraction of the training set's labels are randomly flipped; the test set remains unmodified (i.e., clean). To ensure valid comparison to prior work, which often utilizes architectures specialized to the noisy-label setting, we train a simple model of similar size (ResNet-32) for 200 epochs, following Jiang et al. (2019). We evaluate five variants of model training: standard SGD, SGD with Mixup (Zhang et al., 2017), SAM, and "bootstrapped" variants of SGD with Mixup and SAM (wherein the model is first trained as usual and then retrained from scratch on the labels predicted by the initially trained model). When apply-

| Method | Noise rate (%) | | | |
|---|---|---|---|---|
| | 20 | 40 | 60 | 80 |
| Sanchez et al. (2019) | 94.0 | 92.8 | 90.3 | 74.1 |
| Zhang & Sabuncu (2018) | 89.7 | 87.6 | 82.7 | 67.9 |
| Lee et al. (2019) | 87.1 | 81.8 | 75.4 | - |
| Chen et al. (2019) | 89.7 | - | - | 52.3 |
| Huang et al. (2019) | 92.6 | 90.3 | 43.4 | - |
| MentorNet (2017) | 92.0 | 91.2 | 74.2 | 60.0 |
| Mixup (2017) | 94.0 | 91.5 | 86.8 | 76.9 |
| MentorMix (2019) | **95.6** | **94.2** | 91.3 | **81.0** |
| SGD | 84.8 | 68.8 | 48.2 | 26.2 |
| Mixup | 93.0 | 90.0 | 83.8 | 70.2 |
| Bootstrap + Mixup | 93.3 | 92.0 | 87.6 | 72.0 |
| SAM | 95.1 | 93.4 | 90.5 | 77.9 |
| Bootstrap + SAM | 95.4 | **94.2** | **91.8** | 79.9 |

Table 4: Test accuracy on the clean test set for models trained on CIFAR-10 with noisy labels. Lower block is our implementation, upper block gives scores from the literature, per Jiang et al. (2019).

ing SAM, we use $\rho = 0.1$ for all noise levels except 80%, for which we use $\rho = 0.05$ for more stable convergence. For the Mixup baselines, we tried all values of $\alpha \in \{1, 8, 16, 32\}$ and conservatively report the best score for each noise level.

As seen in Table 4, SAM provides a high degree of robustness to label noise, on par with that provided by state-of-the art procedures that specifically target learning with noisy labels. Indeed, simply training a model with SAM outperforms all prior methods specifically targeting label noise robustness, with the exception of MentorMix (Jiang et al., 2019). However, simply bootstrapping SAM yields performance comparable to that of MentorMix (which is substantially more complex).

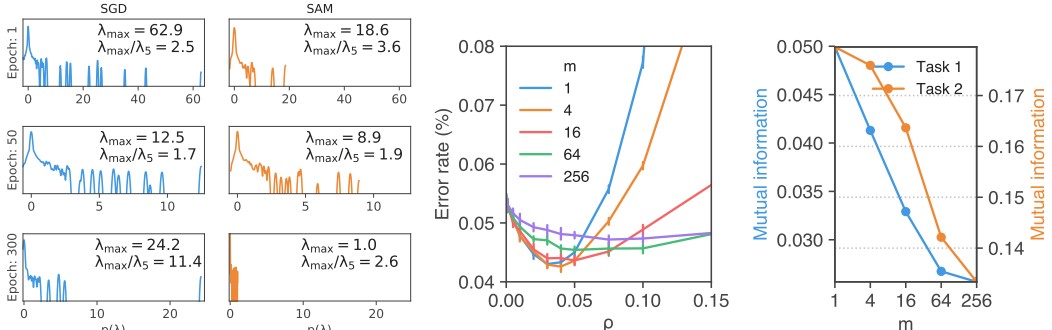

Figure 3: (left) Evolution of the spectrum of the Hessian during training of a model with standard SGD (lefthand column) or SAM (righthand column). (middle) Test error as a function of $\rho$ for different values of $m$. (right) Predictive power of $m$-sharpness for the generalization gap, for different values of $m$ (higher means the sharpness measure is more correlated with actual generalization gap).

## 4 SHARPNESS AND GENERALIZATION THROUGH THE LENS OF SAM

### 4.1 $m$-SHARPNESS

Though our derivation of SAM defines the SAM objective over the entire training set, when utilizing SAM in practice, we compute the SAM update per-batch (as described in Algorithm 1) or even by averaging SAM updates computed independently per-accelerator (where each accelerator receives a subset of size $m$ of a batch, as described in Section 3). This latter setting is equivalent to modifying the SAM objective (equation 1) to sum over a set of independent $\epsilon$ maximizations, each performed on a sum of per-data-point losses on a disjoint subset of $m$ data points, rather than performing the $\epsilon$ maximization over a global sum over the training set (which would be equivalent to setting $m$ to the total training set size). We term the associated measure of sharpness of the loss landscape $m$-*sharpness*.

To better understand the effect of $m$ on SAM, we train a small ResNet on CIFAR-10 using SAM with a range of values of $m$. As seen in Figure 3 (middle), smaller values of $m$ tend to yield models having better generalization ability. This relationship fortuitously aligns with the need to parallelize across multiple accelerators in order to scale training for many of today's models.

Intriguingly, the $m$-sharpness measure described above furthermore exhibits better correlation with models' actual generalization gaps as $m$ decreases, as demonstrated by Figure 3 (right)[7]. In particular, this implies that $m$-sharpness with $m < n$ yields a better predictor of generalization than the full-training-set measure suggested by Theorem 1 in Section 2 above, suggesting an interesting new avenue of future work for understanding generalization.

### 4.2 HESSIAN SPECTRA

Motivated by the connection between geometry of the loss landscape and generalization, we constructed SAM to seek out minima of the training loss landscape having both low loss value and low curvature (i.e., low sharpness). To further confirm that SAM does in fact find minima having low curvature, we compute the spectrum of the Hessian for a WideResNet40-10 trained on CIFAR-10 for 300 steps both with and without SAM (without batch norm, which tends to obscure interpretation of the Hessian), at different epochs during training. Due to the parameter space's dimensionality, we approximate the Hessian spectrum using the Lanczos algorithm of Ghorbani et al. (2019).

Figure 3 (left) reports the resulting Hessian spectra. As expected, the models trained with SAM converge to minima having lower curvature, as seen in the overall distribution of eigenvalues, the

---

[7]We follow the rigorous framework of Jiang et al. (2019), reporting the mutual information between the $m$-sharpness measure and generalization on the two publicly available tasks from the *Predicting generalization in deep learning* NeurIPS2020 competition. https://competitions.codalab.org/competitions/25301

maximum eigenvalue ($\lambda_{\max}$) at convergence (approximately 24 without SAM, 1.0 with SAM), and the bulk of the spectrum (the ratio $\lambda_{\max}/\lambda_5$, commonly used as a proxy for sharpness (Jastrzebski et al., 2020); up to 11.4 without SAM, and 2.6 with SAM).

## 5 RELATED WORK

The idea of searching for "flat" minima can be traced back to Hochreiter & Schmidhuber (1995), and its connection to generalization has seen significant study (Shirish Keskar et al., 2016; Dziugaite & Roy, 2017; Neyshabur et al., 2017; Dinh et al., 2017). In a recent large scale empirical study, Jiang et al. (2019) studied 40 complexity measures and showed that a sharpness-based measure has highest correlation with generalization, which motivates penalizing sharpness. Hochreiter & Schmidhuber (1997) was perhaps the first paper on penalizing the sharpness, regularizing a notion related to Minimum Description Length (MDL). Other ideas which also penalize sharp minima include operating on diffused loss landscape (Mobahi, 2016) and regularizing local entropy (Chaudhari et al., 2016). Another direction is to not penalize the sharpness explicitly, but rather average weights during training; Izmailov et al. (2018) showed that doing so can yield flatter minima that can also generalize better. However, the measures of sharpness proposed previously are difficult to compute and differentiate through. In contrast, SAM is highly scalable as it only needs two gradient computations per iteration. The concurrent work of Sun et al. (2020) focuses on resilience to random and adversarial corruption to expose a model's vulnerabilities; this work is perhaps closest to ours. Our work has a different basis: we develop SAM motivated by a principled starting point in generalization, clearly demonstrate SAM's efficacy via rigorous large-scale empirical evaluation, and surface important practical and theoretical facets of the procedure (e.g., $m$-sharpness). The notion of all-layer margin introduced by Wei & Ma (2020) is closely related to this work; one is adversarial perturbation over the activations of a network and the other over its weights, and there is some coupling between these two quantities.

## 6 DISCUSSION AND FUTURE WORK

In this work, we have introduced SAM, a novel algorithm that improves generalization by simultaneously minimizing loss value and loss sharpness; we have demonstrated SAM's efficacy through a rigorous large-scale empirical evaluation. We have surfaced a number of interesting avenues for future work. On the theoretical side, the notion of per-data-point sharpness yielded by $m$-sharpness (in contrast to global sharpness computed over the entire training set, as has typically been studied in the past) suggests an interesting new lens through which to study generalization. Methodologically, our results suggest that SAM could potentially be used in place of Mixup in robust or semi-supervised methods that currently rely on Mixup (giving, for instance, MentorSAM). We leave to future work a more in-depth investigation of these possibilities.

## 7 ACKNOWLEDGMENTS

We thank our colleagues at Google — Atish Agarwala, Xavier Garcia, Dustin Tran, Yiding Jiang, Basil Mustafa, Samy Bengio — for their feedback and insightful discussions. We also thank the JAX and FLAX teams for going above and beyond to support our implementation. We are grateful to Sven Gowal for his help in replicating EfficientNet using JAX, and Justin Gilmer for his implementation of the Lanczos algorithm[8] used to generate the Hessian spectra. We thank Niru Maheswaranathan for his matplotlib mastery. We also thank David Samuel for providing a PyTorch implementation of SAM[9].

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

## A  APPENDIX

### A.1  PAC BAYESIAN GENERALIZATION BOUND

Below, we state a generalization bound based on sharpness.

**Theorem 2.** *For any $\rho > 0$ and any distribution $\mathscr{D}$, with probability $1 - \delta$ over the choice of the training set $\mathcal{S} \sim \mathscr{D}$,*

$$L_{\mathscr{D}}(\boldsymbol{w}) \leq \max_{\|\boldsymbol{\epsilon}\|_2 \leq \rho} L_{\mathcal{S}}(\boldsymbol{w} + \boldsymbol{\epsilon}) + \sqrt{\frac{k \log\left(1 + \frac{\|\boldsymbol{w}\|_2^2}{\rho^2}\left(1 + \sqrt{\frac{\log(n)}{k}}\right)^2\right) + 4 \log \frac{n}{\delta} + \tilde{O}(1)}{n - 1}} \quad (4)$$

*where $n = |\mathcal{S}|$, $k$ is the number of parameters and we assumed $L_{\mathscr{D}}(\boldsymbol{w}) \leq \mathbb{E}_{\epsilon_i \sim \mathcal{N}(0,\rho)}[L_{\mathscr{D}}(\boldsymbol{w} + \boldsymbol{\epsilon})]$.*

The condition $L_{\mathscr{D}}(\boldsymbol{w}) \leq \mathbb{E}_{\epsilon_i \sim \mathcal{N}(0,\rho)}[L_{\mathscr{D}}(\boldsymbol{w} + \boldsymbol{\epsilon})]$ means that adding Gaussian perturbation should not decrease the test error. This is expected to hold in practice for the final solution but does not necessarily hold for any $\boldsymbol{w}$.

*Proof.* First, note that the right hand side of the bound in the theorem statement is lower bounded by $\sqrt{k \log(1 + \|\boldsymbol{w}\|_2^2/\rho^2)/(4n)}$ which is greater than 1 when $\|\boldsymbol{w}\|_2^2 > \rho^2(\exp(4n/k) - 1)$. In that case, the right hand side becomes greater than 1 in which case the inequality holds trivially. Therefore, in the rest of the proof, we only consider the case when $\|\boldsymbol{w}\|_2^2 \leq \rho^2(\exp(4n/k) - 1)$.

The proof technique we use here is inspired from Chatterji et al. (2020). Using PAC-Bayesian generalization bound McAllester (1999) and following Dziugaite & Roy (2017), the following generalization bound holds for any prior $\mathscr{P}$ over parameters with probability $1 - \delta$ over the choice of the training set $\mathcal{S}$, for any posterior $\mathscr{Q}$ over parameters:

$$\mathbb{E}_{\boldsymbol{w} \sim \mathscr{Q}}[L_{\mathscr{D}}(\boldsymbol{w})] \leq \mathbb{E}_{\boldsymbol{w} \sim \mathscr{Q}}[L_{\mathcal{S}}(\boldsymbol{w})] + \sqrt{\frac{KL(\mathscr{Q}||\mathscr{P}) + \log \frac{n}{\delta}}{2(n - 1)}} \quad (5)$$

Moreover, if $\mathscr{P} = \mathcal{N}(\boldsymbol{\mu}_P, \sigma_P^2 \boldsymbol{I})$ and $\mathscr{Q} = \mathcal{N}(\boldsymbol{\mu}_Q, \sigma_Q^2 \boldsymbol{I})$, then the KL divergence can be written as follows:

$$KL(\mathscr{P}||\mathscr{Q}) = \frac{1}{2}\left[\frac{k\sigma_Q^2 + \|\boldsymbol{\mu}_P - \boldsymbol{\mu}_Q\|_2^2}{\sigma_P^2} - k + k \log\left(\frac{\sigma_P^2}{\sigma_Q^2}\right)\right] \quad (6)$$

Given a posterior standard deviation $\sigma_Q$, one could choose a prior standard deviation $\sigma_P$ to minimize the above KL divergence and hence the generalization bound by taking the derivative[10] of the above

---

[10]Despite the nonconvexity of the function here in $\sigma_P^2$, it has a unique stationary point which happens to be its minimizer.

KL with respect to $\sigma_P$ and setting it to zero. We would then have $\sigma_P^{*2} = \sigma_Q^2 + \|\boldsymbol{\mu}_P - \boldsymbol{\mu}_Q\|_2^2/k$. However, since $\sigma_P$ should be chosen before observing the training data $\mathcal{S}$ and $\boldsymbol{\mu}_Q, \sigma_Q$ could depend on $\mathcal{S}$, we are not allowed to optimize $\sigma_P$ in this way. Instead, one can have a set of predefined values for $\sigma_P$ and pick the best one in that set. See Langford & Caruana (2002) for the discussion around this technique. Given fixed $a, b > 0$, let $T = \{c \exp((1-j)/k)|j \in \mathbb{N}\}$ be that predefined set of values for $\sigma_P^2$. If for any $j \in \mathbb{N}$, the above PAC-Bayesian bound holds for $\sigma_P^2 = c \exp((1-j)/k)$ with probability $1 - \delta_j$ with $\delta_j = \frac{6\delta}{\pi^2 j^2}$, then by the union bound, all above bounds hold simultaneously with probability at least $1 - \sum_{j=1}^{\infty} \frac{6\delta}{\pi^2 j^2} = 1 - \delta$.

Let $\sigma_Q = \rho$, $\boldsymbol{\mu}_Q = \boldsymbol{w}$ and $\boldsymbol{\mu}_P = \boldsymbol{0}$. Therefore, we have:

$$\sigma_Q^2 + \|\boldsymbol{\mu}_P - \boldsymbol{\mu}_Q\|_2^2/k \leq \rho^2 + \|\boldsymbol{w}\|_2^2/k \leq \rho^2(1 + \exp(4n/k)) \tag{7}$$

We now consider the bound that corresponds to $j = \lfloor 1 - k \log((\rho^2 + \|\boldsymbol{w}\|_2^2/k)/c) \rfloor$. We can ensure that $j \in \mathbb{N}$ using inequality equation 7 and by setting $c = \rho^2(1 + \exp(4n/k))$. Furthermore, for $\sigma_P^2 = c \exp((1-j)/k)$, we have:

$$\rho^2 + \|\boldsymbol{w}\|_2^2/k \leq \sigma_P^2 \leq \exp(1/k)\left(\rho^2 + \|\boldsymbol{w}\|_2^2/k\right) \tag{8}$$

Therefore, using the above value for $\sigma_P$, KL divergence can be bounded as follows:

$$KL(\mathscr{P}\|\mathscr{Q}) = \frac{1}{2}\left[\frac{k\sigma_Q^2 + \|\boldsymbol{\mu}_P - \boldsymbol{\mu}_Q\|_2^2}{\sigma_P^2} - k + k\log\left(\frac{\sigma_P^2}{\sigma_Q^2}\right)\right] \tag{9}$$

$$\leq \frac{1}{2}\left[\frac{k(\rho^2 + \|\boldsymbol{w}\|_2^2/k)}{\rho^2 + \|\boldsymbol{w}\|_2^2/k} - k + k\log\left(\frac{\exp(1/k)\left(\rho^2 + \|\boldsymbol{w}\|_2^2/k\right)}{\rho^2}\right)\right] \tag{10}$$

$$= \frac{1}{2}\left[k\log\left(\frac{\exp(1/k)\left(\rho^2 + \|\boldsymbol{w}\|_2^2/k\right)}{\rho^2}\right)\right] \tag{11}$$

$$= \frac{1}{2}\left[1 + k\log\left(1 + \frac{\|\boldsymbol{w}\|_2^2}{k\sigma_Q^2}\right)\right] \tag{12}$$

Given the bound that corresponds to $j$ holds with probability $1 - \delta_j$ for $\delta_j = \frac{6\delta}{\pi^2 j^2}$, the log term in the bound can be written as:

$$\log\frac{n}{\delta_j} = \log\frac{n}{\delta} + \log\frac{\pi^2 j^2}{6}$$

$$\leq \log\frac{n}{\delta} + \log\frac{\pi^2 k^2 \log^2(c/(\rho^2 + \|\boldsymbol{w}\|_2^2/k))}{6}$$

$$\leq \log\frac{n}{\delta} + \log\frac{\pi^2 k^2 \log^2(c/\rho^2)}{6}$$

$$\leq \log\frac{n}{\delta} + \log\frac{\pi^2 k^2 \log^2(1 + \exp(4n/k))}{6}$$

$$\leq \log\frac{n}{\delta} + \log\frac{\pi^2 k^2 (2 + 4n/k)^2}{6}$$

$$\leq \log\frac{n}{\delta} + 2\log\left(6n + 3k\right)$$

Therefore, the generalization bound can be written as follows:

$$\mathbb{E}_{\epsilon_i \sim \mathcal{N}(0,\sigma)}[L_{\mathscr{D}}(\boldsymbol{w}+\boldsymbol{\epsilon})] \leq \mathbb{E}_{\epsilon_i \sim \mathcal{N}(0,\sigma)}[L_{\mathcal{S}}(\boldsymbol{w}+\boldsymbol{\epsilon})] + \sqrt{\frac{\frac{1}{4}k\log\left(1 + \frac{\|\boldsymbol{w}\|_2^2}{k\sigma^2}\right) + \frac{1}{4} + \log\frac{n}{\delta} + 2\log\left(6n + 3k\right)}{n - 1}} \tag{13}$$

In the above bound, we have $\epsilon_i \sim \mathcal{N}(0,\sigma)$. Therefore, $\|\boldsymbol{\epsilon}\|_2^2$ has chi-square distribution and by Lemma 1 in Laurent & Massart (2000), we have that for any positive $t$:

$$P(\|\boldsymbol{\epsilon}\|_2^2 - k\sigma^2 \geq 2\sigma^2\sqrt{kt} + 2t\sigma^2) \leq \exp(-t) \tag{14}$$

Therefore, with probability $1 - 1/\sqrt{n}$ we have that:

$$\|\boldsymbol{\epsilon}\|_2^2 \leq \sigma^2 (2\ln(\sqrt{n}) + k + 2\sqrt{k \ln(\sqrt{n})}) \leq \sigma^2 k \left(1 + \sqrt{\frac{\ln(n)}{k}}\right)^2 \leq \rho^2$$

Substituting the above value for $\sigma$ back to the inequality and using theorem's assumption gives us following inequality:

$$L_{\mathscr{D}}(\boldsymbol{w}) \leq (1 - 1/\sqrt{n}) \max_{\|\boldsymbol{\epsilon}\|_2 \leq \rho} L_{\mathcal{S}}(\boldsymbol{w} + \boldsymbol{\epsilon}) + 1/\sqrt{n}$$

$$+ \sqrt{\frac{\frac{1}{4}k \log\left(1 + \frac{\|\boldsymbol{w}\|_2^2}{\rho^2}\left(1 + \sqrt{\frac{\log(n)}{k}}\right)^2\right) + \log\frac{n}{\delta} + 2\log(6n + 3k)}{n-1}}$$

$$\leq \max_{\|\boldsymbol{\epsilon}\|_2 \leq \rho} L_{\mathcal{S}}(\boldsymbol{w} + \boldsymbol{\epsilon}) +$$

$$+ \sqrt{\frac{k \log\left(1 + \frac{\|\boldsymbol{w}\|_2^2}{\rho^2}\left(1 + \sqrt{\frac{\log(n)}{k}}\right)^2\right) + 4\log\frac{n}{\delta} + 8\log(6n + 3k)}{n-1}}$$

$\square$

## B  ADDITIONAL EXPERIMENTAL RESULTS

### B.1  SVHN AND FASHION-MNIST

We report in table 5 results obtained on SVHN and Fashion-MNIST datasets. On these datasets, SAM allows a simple WideResnet to reach or push state-of-the-art accuracy (0.99% error rate for SVHN, 3.59% for Fashion-MNIST).

For SVHN, we used all the available data (73257 digits for training set + 531131 additional samples). For auto-augment, we use the best policy found on this dataset as described in (Cubuk et al., 2018) plus cutout (Devries & Taylor, 2017). For Fashion-MNIST, the auto-augmentation line correspond to cutout only.

Table 5: Results on SVHN and Fashion-MNIST.

| | | SVHN | | Fashion-MNIST | |
|---|---|---|---|---|---|
| Model | Augmentation | SAM | Baseline | SAM | Baseline |
| Wide-ResNet-28-10 | Basic | $1.42_{\pm 0.02}$ | $1.58_{\pm 0.03}$ | $3.98_{\pm 0.05}$ | $4.57_{\pm 0.07}$ |
| Wide-ResNet-28-10 | Auto augment | $\mathbf{0.99}_{\pm 0.01}$ | $1.14_{\pm 0.04}$ | $\mathbf{3.61}_{\pm 0.06}$ | $3.86_{\pm 0.14}$ |
| Shake-Shake (26 2x96d) | Basic | $1.44_{\pm 0.02}$ | $1.58_{\pm 0.05}$ | $3.97_{\pm 0.09}$ | $4.37_{\pm 0.06}$ |
| Shake-Shake (26 2x96d) | Auto augment | $1.07_{\pm 0.02}$ | $1.03_{\pm 0.02}$ | $\mathbf{3.59}_{\pm 0.01}$ | $3.76_{\pm 0.07}$ |

## C  EXPERIMENT DETAILS

### C.1  HYPERPARAMETERS FOR EXPERIMENTS

We report in table 6 the hyper-parameters selected by gridsearch for the CIFAR experiments, and the ones for SVHN and Fashion-MNIST in 7. For CIFAR10, CIFAR100, SVHN and Fashion-MNIST, we use a batch size of 256 and determine the learning rate and weight decay used to train each model via a joint grid search prior to applying SAM; all other model hyperparameter values are identical to those used in prior work.

For the Imagenet results (Resnet models), the models are trained for 100, 200 or 400 epochs on Google Cloud TPUv3 32 cores with a batch size of 4096. The initial learning rate is set to 1.0 and

Table 6: Hyper-parameter used to produce the CIFAR-{10,100} results

| CIFAR Dataset | LR | WD | $\rho$ (CIFAR-10) | $\rho$ (CIFAR-100) |
|---|---|---|---|---|
| WRN 28-10 (200 epochs) | 0.1 | 0.0005 | 0.05 | 0.1 |
| WRN 28-10 (1800 epochs) | 0.05 | 0.001 | 0.05 | 0.1 |
| WRN 26-2x6 ShakeShake | 0.02 | 0.0010 | 0.02 | 0.05 |
| Pyramid vanilla | 0.05 | 0.0005 | 0.05 | 0.2 |
| Pyramid ShakeDrop (CIFAR-10) | 0.02 | 0.0005 | 0.05 | - |
| Pyramid ShakeDrop (CIFAR-100) | 0.05 | 0.0005 | - | 0.05 |

Table 7: Hyper-parameter used to produce the SVHN and Fashion-MNIST results

| | | LR | WD | $\rho$ |
|---|---|---|---|---|
| SVHN | WRN | 0.01 | 0.0005 | 0.01 |
| | ShakeShake | 0.01 | 0.0005 | 0.01 |
| Fashion | WRN | 0.1 | 0.0005 | 0.05 |
| | ShakeShake | 0.1 | 0.0005 | 0.02 |

decayed using a cosine schedule. Weight decay is set to 0.0001 with SGD optimizer and momentum = 0.9.

Finally, for the noisy label experiments, we also found $\rho$ by gridsearch, computing the accuracy on a (non-noisy) validation set composed of a random subset of 10% of the usual CIFAR training samples. We report the validation accuracy of the bootstrapped version of SAM for different levels of noise and different $\rho$ in table 8.

| | 20% | 40% | 60% | 80% |
|---|---|---|---|---|
| 0 | 15.0% | 31.2% | 52.3% | 73.5% |
| 0.01 | 13.7% | 28.7% | 50.1% | 72.9% |
| 0.02 | 12.8% | 27.8% | 48.9% | 73.1% |
| 0.05 | 11.6% | 25.6% | 47.1% | **21.0**% |
| 0.1 | **4.6**% | **6.0**% | **8.7**% | 56.1% |
| 0.2 | 5.3% | 7.4% | 23.3% | 77.1% |
| 0.5 | 17.6% | 40.9% | 80.1% | 89.9% |

Table 8: Validation accuracy of the bootstrapped-SAM for different levels of noise and different $\rho$

## C.2 FINETUNING DETAILS

Weights are initialized to the values provided by the publicly available checkpoints, except the last dense layer, which change size to accomodate the new number of classes, that is randomly initialized. We train all models with weight decay $1e^{-5}$ as suggested in (Tan & Le, 2019), but we reduce the learning rate to 0.016 as the models tend to diverge for higher values. We use a batch size of 1024 on Google Cloud TPUv3 64 cores and cosine learning rate decay. Because other works train with batch size of 256, we train for 5k steps instead of 20k. We freeze the batch norm statistics and use them for normalization, effectively using the batch norm as we would at test time [11]. We train the models using SGD with momentum 0.9 and cosine learning rate decay. For Efficientnet-L2, we use this time a batch size 512 to save memory and adjusted the number of training steps accordingly. For CIFAR, we use the same autoaugment policy as in the previous experiments. We do not use data augmentation for the other datasets, applying the same preprocessing as for the Imagenet experiments. We also scale down the learning rate to 0.008 as the batch size is now twice as small. We used Google Cloud TPUv3 128 cores. All other parameters stay the same. For Imagenet, we trained both models from checkpoint for 10 epochs using a learning rate of 0.1 and $\rho = 0.05$. We do not randomly initialize the last layer as we did for the other datasets, but instead use the weights included in the checkpoint.

---

[11] We found anecdotal evidence that this makes the finetuning more robust to overtraining.

## C.3 EXPERIMENTAL RESULTS WITH $\rho = 0.05$

A big sensitivity to the choice of hyper-parameters would make a method less easy to use. To demonstrate that SAM performs even when $\rho$ is not finely tuned, we compiled the table for the CIFAR and the finetuning experiments using $\rho = 0.05$. Please note that we already used $\rho = 0.05$ for all Imagenet experiments. We report those scores in table 9 and 10.

| Model | Augmentation | Cifar10 | | Cifar100 | |
|---|---|---|---|---|---|
| | | $\rho = 0.05$ | SGD | rho=0.05 | SGD |
| WRN-28-10 (200 epochs) | Basic | **2.7** | 3.5 | **16.5** | 18.8 |
| WRN-28-10 (200 epochs) | Cutout | **2.3** | 2.6 | **14.9** | 16.9 |
| WRN-28-10 (200 epochs) | AA | **2.1** | 2.3 | **13.6** | 15.8 |
| WRN-28-10 (1800 epochs) | Basic | **2.4** | 3.5 | **16.3** | 19.1 |
| WRN-28-10 (1800 epochs) | Cutout | **2.1** | 2.7 | **14.0** | 17.4 |
| WRN-28-10 (1800 epochs) | AA | **1.6** | 2.2 | **12.8** | 16.1 |
| WRN 26-2x6 ss | Basic | **2.4** | 2.7 | **15.1** | 17.0 |
| WRN 26-2x6 ss | Cutout | **2.0** | 2.3 | **14.2** | 15.7 |
| WRN 26-2x6 ss | AA | **1.7** | 1.9 | **12.8** | 14.1 |
| PyramidNet | Basic | **2.1** | 4.0 | **15.4** | 19.7 |
| PyramidNet | Cutout | **1.6** | 2.5 | **13.1** | 16.4 |
| PyramidNet | AA | **1.4** | 1.9 | **12.1** | 14.6 |
| PyramidNet+ShakeDrop | Basic | **2.1** | 2.5 | **13.3** | 14.5 |
| PyramidNet+ShakeDrop | Cutout | **1.6** | 1.9 | **11.3** | 11.8 |
| PyramidNet+ShakeDrop | AA | **1.4** | 1.6 | **10.3** | 10.6 |

Table 9: Results for the Cifar10/Cifar100 experiments, using $\rho = 0.05$ for all models/datasets/augmentations

| Dataset | Efficientnet-b7 + SAM (optimal) | Efficientnet-b7 + SAM ($\rho = 0.05$) | Efficientnet-b7 |
|---|---|---|---|
| FGVC_Aircraft | 6.80 | 7.06 | 8.15 |
| Flowers | 0.63 | 0.81 | 1.16 |
| Oxford_IIIT_Pets | 3.97 | 4.15 | 4.24 |
| Stanford_Cars | 5.18 | 5.57 | 5.94 |
| cifar10 | 0.88 | 0.88 | 0.95 |
| cifar100 | 7.44 | 7.56 | 7.68 |
| Birdsnap | 13.64 | 13.64 | 14.30 |
| Food101 | 7.02 | 7.06 | 7.17 |

Table 10: Results for the the finetuning experiments, using $\rho = 0.05$ for all datasets.

## C.4 ABLATION OF THE SECOND ORDER TERMS

As described in section 2, computing the gradient of the sharpness aware objective yield some second order terms that are more expensive to compute. To analyze this ablation more in depth, we trained a Wideresnet-40x2 on CIFAR-10 using SAM with and without discarding the second order terms during training. We report the cosine similarity of the two updates in figure 5, along the training trajectory of both experiments. We also report the training error rate (evaluated at $\boldsymbol{w} + \hat{\boldsymbol{\epsilon}}(\boldsymbol{w})$) and the test error rate (evaluated at $\boldsymbol{w}$).

We observe that during the first half of the training, discarding the second order terms does not impact the general direction of the training, as the cosine similarity between the first and second order updates are very close to 1. However, when the model nears convergence, the similarity between both types of updates becomes weaker. Fortunately, the model trained without the second order terms reaches a lower test error, showing that the most efficient method is also the one providing the best generalization on this example. The reason for this is quite unclear and should be analyzed in follow up work.

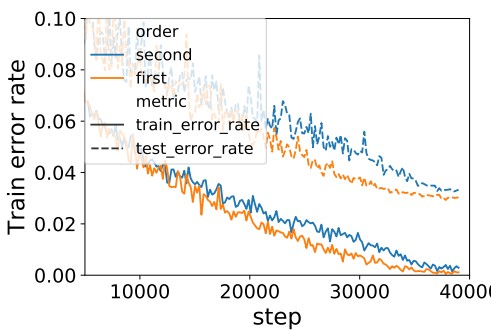 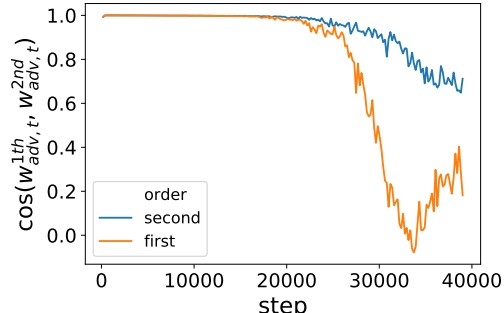

Figure 4: Training and test error for the first and second order version of the algorithm.

Figure 5: Cosine similarity between the first and second order updates.

### C.5 CHOICE OF P-NORM

Our theorem is derived for $p = 2$, although generalizations can be considered for $p \in [1, +\infty]$ (the expression of the bound becoming way more involved). Empirically, we validate that the choice $p = 2$ is optimal by training a wide resnet on cifar10 with SAM for $p = \infty$ (in which case we have $\hat{\epsilon}(\boldsymbol{w}) = \rho \, \mathrm{sign}\left(\nabla_{\boldsymbol{w}} L_{\mathcal{S}}(\boldsymbol{w})\right)$) and $p = 2$ (giving $\hat{\epsilon}(\boldsymbol{w}) = \frac{\rho}{||\nabla_{\boldsymbol{w}} L_{\mathcal{S}}(\boldsymbol{w})||_2^2}(\nabla_{\boldsymbol{w}} L_{\mathcal{S}}(\boldsymbol{w}))$). We do not consider the case $p = 1$ which would give us a perturbation on a single weight. As an additional ablation study, we also use random weight perturbations of a fixed Euclidean norm: $\hat{\epsilon}(\boldsymbol{w}) = \frac{\rho}{||\boldsymbol{z}||_2^2}\boldsymbol{z}$ with $\boldsymbol{z} \sim \mathcal{N}(\boldsymbol{0}, \boldsymbol{I}_d)$. We report the test accuracy of the model in figure 6.

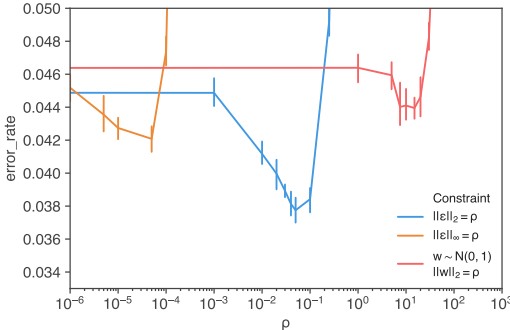

Figure 6: Test accuracy for a wide resnet trained on CIFAR10 with SAM, for different perturbation norms.

We observe that adversarial perturbations outperform random perturbations, and that using $p = 2$ yield superior accuracy on this example.

### C.6 SEVERAL ITERATIONS IN THE INNER MAXIMIZATION

To empirically verify that the linearization of the inner problem is sensible, we trained a WideResnet on the CIFAR datasets using a variant of SAM that performs several iterations of projected gradient ascent to estimate $\max_\epsilon L(w + \epsilon)$. We report the evolution of $\max_\epsilon L(w + \epsilon) - L(w)$ during training (where $L$ stands for the training error rate computed on the current batch) in Figure 7, along with the test accuracy and the estimated sharpness ($\max_\epsilon L(w + \epsilon) - L(w)$) at the end of training in Table 11; we report means and standard deviations across 20 runs.

For most of the training, one projected gradient step (as used in standard SAM) is sufficient to obtain a good approximation of the $\epsilon$ found with multiple inner maximization steps. We however observe that this approximation becomes weaker near convergence, where doing several iterations of projected gradient ascent yields a better $\epsilon$ (for example, on CIFAR-10, the maximum loss found

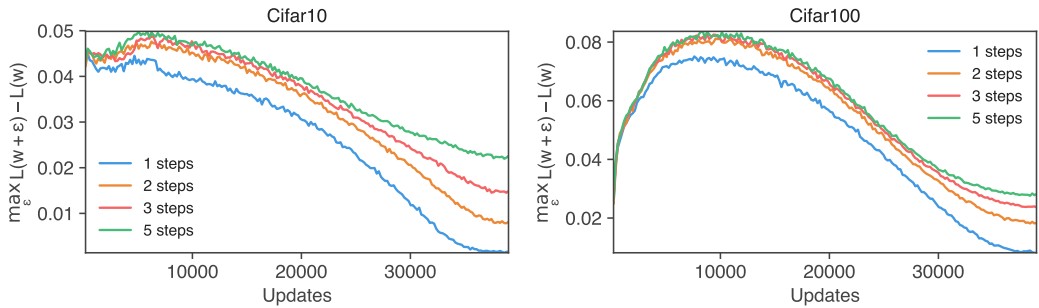

Figure 7: Evolution of $\max_\epsilon L(w + \epsilon) - L(w)$ vs. training step, for different numbers of inner projected gradient steps.

on each batch is about 3% more when doing 5 steps of inner maximization, compared to when doing a single step). That said, as seen in Table 11, the test accuracy is not strongly affected by the number of inner maximization iterations, though on CIFAR-100 it does seem that several steps outperform a single step in a statistically significant way.

| Number of projected | CIFAR-10 | | CIFAR-100 | |
|---|---|---|---|---|
| gradient steps | Test error | Estimated sharpness | Test error | Estimated sharpness |
| 1 | $2.77_{\pm 0.03}$ | $0.17_{\pm 0.03}$ | $16.72_{\pm 0.08}$ | $0.82_{\pm 0.05}$ |
| 2 | $2.76_{\pm 0.03}$ | $0.82_{\pm 0.03}$ | $16.59_{\pm 0.08}$ | $1.83_{\pm 0.05}$ |
| 3 | $2.73_{\pm 0.04}$ | $1.49_{\pm 0.05}$ | $16.62_{\pm 0.09}$ | $2.36_{\pm 0.03}$ |
| 5 | $2.77_{\pm 0.03}$ | $2.26_{\pm 0.05}$ | $16.60_{\pm 0.06}$ | $2.82_{\pm 0.04}$ |

Table 11: Test error rate and estimated sharpness ($\max_\epsilon L(w + \epsilon) - L(w)$) at the end of the training.

