# OpenReview forum: "Sharpness-aware Minimization for Efficiently Improving Generalization"
_ICLR.cc/2021/Conference — ICLR 2021 Spotlight_

### Official Review · AnonReviewer2 · 2020-10-22
**Good paper, but considerable room for improvement**

**Rating:** 8
**Confidence:** 4

**Review:**

## Summary

This paper proposes and empirically evaluates SAM, an optimization method that is designed to seek out regions of uniformly low training loss. The method is derived from a bound on the generalization performance of parameters $w$ in terms of the maximal training loss in a region around $w$. After various approximations, minimizing this upper bound gives rise to a simple method, which first performs a (normalized) gradient ascent step; computes the gradient at that perturbed location; and uses that gradient to update the weights. The empirical evaluation shows that SAM improves generalization performance across a wide range of settings.

## Rating

The paper is well-structured and easy to read. The proposed method is motivated from a PAC-Bayesian generalization bound, but involves approximations which are only vaguely justified. The empirical evaluation is extensive and rigorous. Given its simplicity, scalability, and the convincing results, I believe that this method could have significant impact. With a small exception (see below) all derivations are, to the best of my knowledge, mathematically sound. Overall, this is a good paper and I recommend acceptance, but there are various aspects that could be improved. If these are addressed during the rebuttal, I will consider increasing my score.

## Major Comments

1) The proposed method is motivated by a generalization bound, but this bound is never evaluated numerically. Prior work has shown that many generalization bounds are numerically vacuous in deep learning settings. The bound in Theorem 1 should be really straight-forward to evaluate after the end of training and I think this would be a vital addition to this paper. The paper shows convincing practical benefits of SAM, but it is important to know whether this benefit actually stems from the (approximate) minimization of a non-vacuous bound. This is my single biggest concern with this work.

2) The derivation of the SAM update relies on various approximations to turn the intractable bi-level optimization problem of Eq. (1) into a practical method. My concern here is that the paper does not even attempt to justify these approximations and/or to discuss their limitations and possible pitfalls. I’m not saying that these approximations are not justified; I just think that a bit of discussion would significantly strengthen the paper. A few thoughts:
    a) The linear approximation that facilitates the closed-form solution of the inner optimization problem (top of page 4) is really drastic. It would be accurate if and only if the curvature in a $\rho$-sized region around w would be negligibly small. That, however, is exactly the type of region the method wants to find, so this is somewhat circular. Contrasting the first-order with a second-order approximation may be a useful exercise. An interesting ablation would be to solve the inner loop optimization in Eq. (1) more accurately, say, by performing multiple iterations of (projected) gradient descent. (When dropping the second-order term, this wouldn’t be exceedingly costly and could provide further insight.)
    b) Later on, the “second order” term, which derives through the solution $\hat{\epsilon}$ of the inner-loop optimization, is dropped. The authors write that this is done to “further accelerate the computation” and “does not adversely affect model training and in fact improves numerical stability”. However, as shown in Appendix C.3, dropping the second-order term yields a quite considerable *improvement*. It always makes me a bit nervous, when a (relatively crude) approximation works better than the quantity you actually want to compute. I don’t think it is adequate to sweep this under the rug with a vague reference to numerical stability. At least, I thank that (i) the fact should be stated clearly in the main text, and (ii) the reference to numerical stability should be marked as speculation, unless there’s evidence to back it up.

3) Even if the approximations involved are only vaguely justified, the SAM update itself makes a lot of intuitive sense, and I think the reader might benefit from discussing that in a bit more detail. For example, one simple observation would be that, for a sharp minimum whose basin of attraction is of radius smaller than rho, the ascent step of SAM (by virtue of the normalisation to length rho) would take you outside of that basin. If the basin of attraction is larger than rho, the SAM update stays inside. Maybe this intuition could be formalized?

4) Connection to Entropy-SGD:
    a) Is it really fair to characterize EntropySGD as “only suitable to small models and datasets”? Of course, it requires a number L of additional gradient steps to obtain an MC-estimate of the gradient of the local entropy. But, if my understanding is correct, these could be parallelized. The original paper uses L=20 but, to my knowledge, nobody has tested the (lower) limits of that number. If L<=5 does something useful, it wouldn’t be too outlandish in terms of computational cost.
    b) On that note: It would have been great to see an empirical comparison to EntropySGD for at least one of the experiments.
    c) Looking more closely at the proof of the generalization bound, the term involving maximization over a local region actually originates from a bound on the *integral* over that region, which makes the connection to EntropySGD even closer. Is it fair to say that the maximization step in SAM is actually just a clever and cheap surrogate for integrating the gradient the over the local region as done in EntropySGD?
    d) An interesting piece of related work that you might want to add is [1], which shows that EntropySGD (in a specific sense) also optimizes a generalization bound.

5) You clearly show the benefits of SAM in pushing the state-of-the-art, which is fantastic! Personally, I would have also valued experiments that are more illustrative and maybe test the limits of the method:
    a) What does SAM do in a convex setting, say, a logistic regression on some (maybe synthetic) dataset that is easily overfit? Does it converge to the unique minimizer of the training loss? Does it plateau before reaching that?
    b) What happens if you apply SAM in the setting of [2] with random labels assigned to images? Will it converge to a point that closely fits each (nonsense) training sample?
    c) With regards to the finding on m-sharpness, it would be great to see a small-scale experiments with full-batch optimization.

6) The finding on m-sharpness is intriguing, though somewhat counter to the motivation for SAM laid out in the paper. The results in Figure 3 make me wonder, whether SAM would even gives practical improvements with large m? This is an important question, but I totally agree with the authors decision to leave it for future work.

## Minor Comments

7) SAM is motivated from the generalization bound in Theorem 1. The bound includes a term $h(||w||^2 / \rho^2)$, which is later replaced with a simple L2 regularization term with coefficient lambda. From the Theorem, that coefficient $\lambda$ should be coupled with the neighborhood size $\rho$ in an inversely proportional manner. Of course, for a fixed choice of $\rho$, this can be subsumed into the regularization parameter lambda. However, it would make a difference when comparing various choices for rho, as is done in the experiments. Did you keep the regularization parameter lambda constant in those experiments or did it vary with rho, as suggested by the Theorem? Furthermore, it would be interesting to know to what extent SAM depends on that L2 regularization. Did all experiments use L2 regularization and did you, by any chance, test SAM’s performance without the regularization parameter?

8) The bib file could really need some love:
    a) You cite arXiv versions of of multiple papers that have been published at peer-reviewed venues.
    b) Capitalize properly: “nesterov momentum”, “pac-bayesian”, “journal of machine learning research”, etc…
    c) Cite identical venues consistently.

9) An interesting connection (or non-connection) to discuss in the paper would be adversarial training. Some of these methods perform an adversarial perturbation step to the *data*, followed by a weight update computed on that perturbed data. This is an interesting parallel to SAM, where the first step is sort of an adversarial perturbation of the weights.

10) If I am not mistaken, there is a minor flaw in the proof of Theorem 1. The very last equation gives a high-probability bound on the norm of $||\epsilon||^2$, which is then plugged into the high-probability generalization bound of Eq. (). Both bounds hold with probability 1-delta individually, but that does not imply that the combined bound holds with probability 1-delta. However, that should easily be fixed with a slight adaptation of the constants.

## Typos / Style

- At least according to some style guides, I think that you should write “Sharpness-Aware Minimization” instead of “Sharpness-aware Minimization”.
- Capitalize references to sections, equations, figures: “in Appendix C.3”, “using Equation 3”, et cetera…
- Typo “opefrations” near the bottom of page 4
- Footnote markers that refer to an entire sentence go after the punctuation mark (e.g., for footnotes 3 and 5).
- In Table 3, the boldface is assigned to the wrong method in the Stanford Cars row.

## References

[1] Dziugaite, G. K., & Roy, D. (2018). Entropy-SGD optimizes the prior of a PAC-Bayes bound: Generalization properties of Entropy-SGD and data-dependent priors. In International Conference on Machine Learning (pp. 1377-1386). PMLR.

[2] Zhang, C., Bengio, S., Hardt, M., Recht, B., & Vinyals, O. (2016). Understanding deep learning requires rethinking generalization. arXiv preprint arXiv:1611.03530.

## Update after Rebuttal

Thanks for the detailed replies to my questions and comments. I think the paper has been improved substantially and I have increased my rating. Congratulations on the good work!

---

> ### Author Response · Authors · 2020-11-17
> **Thanks for your detailed engagement! We have done our best to address your concerns. (part 3)**
>
> **What happens if you apply SAM in the setting of [2] with random labels assigned to images? Will it converge to a point that closely fits each (nonsense) training sample?** Our experiments from section 3.3 explore this (though not using 100% random labels, which could also be interesting), although in the paper we only report the test accuracy. With respect to training accuracy, what we observe is that when using SAM, the network interpolates the “clean” part of the training dataset, while not learning the corrupted labels (it is also worth noting that the same phenomenon can be obtained with other regularizers, such as Mixup with high alpha). For SAM, this might suggest that the minima that fit random labels have a smaller width than the minima that fit the true labels, although this is speculative at this point and we decided to leave this out of the final manuscript.
> We however believe that there is some very interesting research to be done on the intersection between SAM and semi-supervised / noisy or random labels, and that this could provide more insight into the geometry of the loss landscape.
>
> **With regards to the finding on m-sharpness, it would be great to see a small-scale experiments with full-batch optimization.** Given the results of the m-sharpness experiment, and the fact that non stochastic gradient descent performs poorly for deep networks, we believe that full batch optimization would not perform well. It would be interesting to see how much SAM manages to mitigate the generalization issue of full batch gradient descent though.
>
> **The finding on m-sharpness is intriguing, though somewhat counter to the motivation for SAM laid out in the paper.** We initially thought the same thing as the reviewer, as it seemed that branching away from the definition of sharpness used in the theorem (m=dataset size) makes the connection between SAM and the generalization bound more tenuous. However, we now believe that the results presented in the right side of figure 3 (Predictive power of m-sharpness for the generalization gap) changes this perspective, and shows that there might be better bounds to be discovered that rely on a “per example sharpness”. In that sense, SAM could be an argument to explore new types of sharpness-based bounds.
>
> **From the Theorem, that coefficient λ should be coupled with the neighborhood size ρ in an inversely proportional manner.** This is a great point. In our experiments, we wanted to be i) mimicking the way most practitioners would tune the parameters and ii) being conservative in our choice of rho, avoiding large gridsearch on rho x lambda x learning rate for instance. As a result, we first tune the baseline to find the optimal learning rate and the weight decay (jointly), and only after that tune rho when using SAM. It is however true that we could have used lambda/rho instead of lambda when searching for rho, and it would be interesting to see the impact of this on the results.
>
> **An interesting connection (or non-connection) to discuss in the paper would be adversarial training.** Some of these methods perform an adversarial perturbation step to the data -> SAM can indeed been described as a parameter space analogue to the fast gradient sign method (FGSM) (although the FGSM  projection is done on the l_inf sphere and not the l_2 sphere), as they both solve min max problems. It would be interesting to see how networks performs when mixing both, considering adversarial attacks on both the input and the weights, but we leave this to  future work.
>
> **The bib file could really need some love.** You are correct, we will clean it in the next version of the paper.
>
> **If I am not mistaken, there is a minor flaw in the proof of Theorem 1.** We added a corrected version of the theorem.
>
> **In Table 3, the boldface is assigned to the wrong method in the Stanford Cars row.**  The boldface is actually correct, it is the score that should read 5.0 instead of 5.3 in this version of the paper. We will correct the table.
>
>
>
> References:
>
> [1] Exploring the Vulnerability of Deep Neural Networks: A Study of Parameter Corruption (https://arxiv.org/pdf/2006.05620.pdf)
>
> [2]Regularizing Neural Networks via Adversarial Model Perturbation (https://arxiv.org/abs/2010.04925) <- maybe not needed
>
> [3] AdaNet: Adaptive Structural Learning of Artificial Neural Networks,

---

> > ### Comment · AnonReviewer2 · 2020-11-18
> > **Thanks for the detailed response**
> >
> > Thanks for the detailed reply and for your efforts to address my comments. You have answered many of the questions I had; please consider comments that I’m not responding to as resolved. However, my two main concerns remain. Please understand that these are mostly not about the substance of the paper, but rather about the presentation. Nevertheless, I find them to be very important.
> >
> >
> > **The proposed method is motivated by a generalization bound, but this bound is never evaluated numerically.**
> >
> > You responded that it is okay *“to propose an algorithm inspired by generalization bounds, even if those bounds are not expected to be tight."* I don’t take issue with a non-tight bound, I take issue with a vacuous bound, because **a vacuous generalization bound is not a generalization bound.** I don’t intend to tell you where to find inspiration, but the way the paper is currently written strongly suggests to the reader that the algorithm is good *because* it minimizes a generalization bound. That doesn’t seem to be the case. (Do you assume that the bound is vacuous or have you checked?) The problem is further exacerbated by the open questions around $m$-sharpness.
> >
> > In light of this, I find the motivation the paper currently gives misleading and I think it should be adjusted. Personally, I would motivate this method along the following lines:
> > - Sharpness is thought to be related to generalization, so we want to have a method that finds flat minima
> > - Prior work has mostly considered average-based notions of sharpness; these are hard to turn into practical algorithms
> > - We consider a maximum-based notion of sharpness, which (together with some approximations) gives rise to a simple and efficient algorithm
> > - We provide an extensive empirical evaluation of the method
> >
> > Of course, you don't want to completely turn the paper on its head in this stage of the process. But I believe that you should adjust the abstract/introduction in order to clarify things.
> >
> >
> > **Concerns about the second order term.**
> >
> > You write that you *“hope that the projected gradient experiment outlined above provides some reassurance, as it shows that a better approximation of SAM loss (via a better epsilon) does not decrease the accuracy.”* I don’t think it does because these are two separate issues. Even if you solve the inner maximization problem exactly that doesn’t in itself justify dropping the second-order term. As I wrote in my review, all I’m asking for is that the paper acknowledge and discuss these limitations / open questions.
> >
> >
> > **Performing multiple iterations of (projected) gradient descent**
> >
> > This is a very interesting addition to the paper. Thanks for your effort!
> >
> >
> > I hope my comments don't seem overly critical. I'm just trying to help improve an already good paper!

---

> > > ### Author Response · Authors · 2020-11-19
> > > **Thanks for your comments. We have addressed both of them.**
> > >
> > > Thanks for your engagement with our response. It appears that our effort has paid off and we have thus far been able to address all of your concerns except two. While we do not fully agree with some aspects of your comments, we have updated the paper addressing both of your concerns by following your suggestions. With that in mind, we respectfully ask that you consider strongly supporting the acceptance of our paper by increasing your score accordingly.
> > >
> > > ### Framing the motivation
> > >
> > > We did not seek to claim in the paper that “the algorithm is good because it minimizes a generalization bound,” nor did we intend to imply that (and we believe that we did not do so).  That said, in order to further eliminate any risk that the paper could be interpreted as suggesting that, we have made the following changes to address your concern:
> > >
> > > 1) We have modified the presentation in the paper to be fully aligned with the narrative that the reviewer suggested.  Notably:
> > >  - We have removed any mention of our generalization bound from the abstract and introduction.
> > >  - We emphasize in the abstract and introduction that prior work showing that there is a connection between sharpness and generalization (and not our generalization bound) provides the motivation for pursuing flat minima.
> > >  - We have added a paragraph to the introduction introducing prior work on flatness and approaches that have sought to find flat minima (and refer the reader to the related work section for a more detailed discussion) to motivate our algorithm based on prior work as the reviewer suggested.
> > >  - We then present SAM as a new algorithm that uses a different approach to minimizing sharpness (along with loss), yielding a simple and efficient algorithm that we show is effective via a rigorous empirical evaluation.
> > >  - We have reviewed our language throughout the paper (including beyond the abstract and introduction), seeking to modify or remove any language that could potentially be at all interpreted as suggesting that “the algorithm is good because it minimizes a generalization bound.”  For example, we have reworded our introduction of the algorithm after Theorem 1 to read “Thus, inspired by the terms from the bound, we propose to select parameter values by solving the following Sharpness-Aware Minimization (SAM) problem”, which we believe clearly conveys that the bound serves only as a source of inspiration and not that the algorithm minimizes the bound (and in turn certainly not that the algorithm is good because it minimizes a generalization bound).
> > >
> > > We believe that we have successfully addressed your comments as we followed your suggestions in the previous comment. If there is any other part of the paper that you feel would imply that “the algorithm is good because it minimizes a generalization bound,” please let us know.
> > >
> > > ### Concerns about the second order terms:
> > >
> > > As suggested by the reviewer, we have “acknowledge[d] and discuss[ed] [the] limitations / open questions” around not including the second-order terms by adding text doing so immediately after Equation 3.  We now explicitly state that our primary validation for performance in the absence of the second-order terms is the empirical results in Section 3 (which show that the resulting algorithm is unequivocally effective without the second-order terms).  We go on to illustrate that we nonetheless do not yet fully understand the limitations and/or pitfalls of removing the second-order terms, referring explicitly to the surprising result of our initial experiment in that vein, and stating that further investigating should be a priority in future work.

---

> > > > ### Comment · AnonReviewer2 · 2020-11-19
> > > > **Response**
> > > >
> > > > I've read the updated paper and the adjustments are a step in the right direction, but not as clear as I had hoped. E.g., the paper does not acknowledge that the bound is numerically vacuous, which I think would be important information. Also, unless the reader dives into the details of the proof, they will not realize that the maximization term is merely a surrogate for an average-based notion of sharpness; wouldn't that be interesting to know?
> > > >
> > > > In any case, you have addressed most of the concerns that I had and I will increase my score correspondingly. Congratulations on a good paper!

---

> > > > > ### Author Response · Authors · 2020-11-23
> > > > > **Response**
> > > > >
> > > > > Thank you for your comment and for increasing your score. For the final version:
> > > > >  - We will add the explanation about the nature of the proof.
> > > > >  - We will evaluate the bound numerically and report its value.

---

> ### Author Response · Authors · 2020-11-17
> **Thanks for your detailed engagement! We have done our best to address your concerns. (part 2)**
>
> **Concerns about the second order term:** We agree and are editing the paper to clarify and address this point clearly in the main text; we intend to upload an updated version of the paper shortly. Additionally, we plan to further investigate this behavior; in the meantime we hope that the projected gradient experiment outlined above provides some reassurance, as it shows that a better approximation of SAM loss (via a better epsilon) does not decrease the accuracy.
>
>
> **But, if my understanding is correct, these could be parallelized. The original paper uses L=20 but, to my knowledge, nobody has tested the (lower) limits of that number:** Our understanding is that these updates can not be parallelized (beside the usual data parallelism) and must be done sequentially. The paper describes that the authors tried L between 5 and 20 depending on the dataset, but kept L=20 for even simple image tasks (MNIST and Cifar), suggesting that lower values of L might not perform as well. In their experiment, they however control for the added computations by training the baseline for L time as long (as we do for table 1 when we report the best score of the SGD model trained for n or 2n epochs), but it is unclear how good is the baseline they are comparing against (100 epochs for 1.4% error on MNIST seems excessive). We will replace “However, their proposed algorithm is computationally expensive and is only suitable to small models and datasets” by “However, their proposed algorithm is computationally more expensive than SAM” on the off chance that EntropySGD could indeed be applied to modern, larger nets.
>
> **Is it fair to say that the maximization step in SAM is actually just a clever and cheap surrogate for integrating the gradient the over the local region as done in EntropySGD.** We believe it is indeed fair to call SAM a cheap surrogate of Entropy SGD in some sense. The goal here is to find a measure of sharpness that is easy to compute and differentiate through, and we believe SAM represents an advancement in that regard with respect to other methods such as EntropySGD. The reviewer is correct in pointing out that even the proof of our theorem calls for a “mean” version of sharpness such as the one presented in the entropy SGD paper. Informally, SAM leverages the fact that in high dimension, it is easier to estimate the max (at the cost of a single gradient call) than the “average”, that would have to use some kind of costly MC method (such as the one used in EntropySGD). We also find Dziugaite, G. K., & Roy, D. (2018). Entropy-SGD optimizes the prior of a PAC-Bayes bound very relevant and we will be sure to discuss it in the paper.
>
> **You clearly show the benefits of SAM in pushing the state-of-the-art, which is fantastic.** We thank the reviewer for these encouraging comments!  We’ve also updated the paper with new results obtained by fine tuning efficientnet-l2-475 on imagenet using SAM, yielding a new state of the art on this dataset as well.

---

> ### Author Response · Authors · 2020-11-17
> **Thanks for your detailed engagement! We have done our best to address your concerns. (part 1)**
>
> Thanks for your valuable feedback. We are excited by your detailed engagement with our paper! We have tried our best to address all your concerns in the last few days. We would really appreciate it if you consider increasing your score.
>
> **The proposed method is motivated by a generalization bound, but this bound is never evaluated numerically.** As you noted, the bounds are numerically vacuous for the models we used, as the tools used for the proof are loose. Our algorithm is inspired from the generalization bound and we mainly benefit from the relationship between terms in the generalization bound to formulate our objective function. It is also worth mentioning that our work is not the first to propose an algorithm inspired by generalization bounds, even if those bounds are not expected to be tight (see [3] for instance).
>
> **Is important to know whether this benefit actually stems from the (approximate) minimization of a non-vacuous bound**  Although we agree that the bound is not systematically evaluated in our paper, we believe that figure 3 (middle and right) provides some elements suggesting that the benefits of SAM correlate well with the minimization of the bound. In this experiment, we show that using a smaller m leads to better generalization (middle figure), but also that smaller m leads to a better predictor of the generalization gap (right figure, evaluated on the public datasets from the NeurIPS competition organized by Jiang et al). Although we agree that this is not a definite answer to the concerns of the reviewer, we think this finding is reassuring.
>
> We understand this is your main concern, and thus we are currently running additional experiments to address it. We will update this review with these new elements shortly.
>
> **My concern here is that the paper does not even attempt to justify these approximations and/or to discuss their limitations and possible pitfalls** We agree that it would be useful to further investigate the potential limitations of the SAM algorithm’s approximation to the SAM objective.  Indeed, it is easy to construct L: w -> R such that this approximation is arbitrarily bad, and bounding the approximation’s error would likely require assumptions that realistic settings (e.g., involving non-convex loss) would not satisfy (see for example Theorem 2 from concurrent work [1], which essentially provides the bound the reviewer is asking for but requires convexity and L-smoothness). As a result, we focused on performing a rigorous empirical investigation in order  to show that the approximation is useful, though we agree that developing a better theoretical understanding of its behavior and limitations would be an excellent direction for future work.
>
> **An interesting ablation would be to solve the inner loop optimization in Eq. (1) more accurately, say, by performing multiple iterations of (projected) gradient descent.** We added a new appendix analyzing SAM when several steps of projected gradient ascent are used to compute epsilon. The conclusion of this additional experiment is that i) for most of the training, one projected gradient step is enough to get a good approximation of the epsilon found with several, ii) this approximation becomes weaker near convergence, where doing several iterations of projected gradient ascent yield a better epsilon, and iii) the test accuracy is not strongly affected by the number of projected gradient iterations (when it is affected, increasing the number of projected gradient iterations increases test accuracy slightly).

---

### Official Review · AnonReviewer4 · 2020-10-28
**Clean and simple method, strong empirical results.**

**Rating:** 8
**Confidence:** 3

**Review:**

## Summary ##
The paper proposes a modified loss function for supervised learning, in which the original loss at w is replaced with a maximum of the loss in a small p-norm ball around w. An approximate way to compute gradients for this loss is presented, and evaluated in high detail on a variety of supervised learning problems where it is shown to consistently improve the overall generalization error. Furthermore, based on PAC-Bayes theory, a generalization bound for learning under that loss is presented.

## Explanation of Rating ##
The main strengths of the paper are the simplicity and convincing evaluation of the method. Another strength is the sound theoretical generalization bound. The paper is also very well written, and I therefore clearly recommend acceptance.

Perhaps a weakness of the paper is the lack of a "broader/high-level perspective" on the method (see detailed comment #1) and its comparison to variational inference methods which also optimize PAC-Bayes bounds (see detailed comment #2).

## Detailed Comments ##

1. In the past, Gaussian convolution smoothings of loss functions have been considered a lot in the context of homotopy continuation methods or variational inference. To me, the proposed loss function \max_\eps L(w + \eps) - \delta_{|\eps| < \rho} can be seen as a convolution of the loss, but on the tropical semiring (max, +) with the convolution kernel being the indicator function \delta_{|\eps| < \rho}. Such types of convolution have been heavily studied in the field of convex analysis, where they are known under the name infimal convolution / epi-addition. This opens up the question on why this specific convolution kernel has been chosen. The practical results in this paper are quite strong and have been elusive so far for variational inference and Gaussian smoothing methods, even though they have been around for a long time. Do these good results mainly stem from the efficiently implementable algorithm, or are they more due to favorable geometrical properties of the loss due to performing the convolution in the tropical geometry?

2. What is the main advantage of the proposed approach over a variational inference (VI) method (e.g. with mean-field Gaussian approximation) to minimize the PAC-Bayes bound? In variational inference, the free parameter \rho is also be optimized by minimizing the right-hand side of the PAC-Bayes bound. Have you tried minimizing over \rho, or perhaps consider a diagonal approximation of \rho?

Minor comments / typos:
- p.12 "Adding a Gaussian perturbation should increase the test error" -> "Adding a Gaussian perturbation should not decrease the test error."  This is expected to hold in practice at a minimum. But it is not clear if it holds for any w.
- p.12 Typo: "Then we KL divergence" -> "Then the KL divergence".
- The steps (12) and (13) in the proof are not easy to follow. In particular, it was unclear to me what meant with "each bound" before Eq. 12, and why the bounds should hold with this specific choice of probability. Furthermore, I couldn't follow what happens from (12) to (13).
- I did not notice where the assumption ||w|| >= 1 is used, perhaps it can be mentioned in the proof.
- \lambda is another hyperparameter, but it is not mentioned later on how it is chosen.

---

> ### Author Response · Authors · 2020-11-17
> **Thank you for your encouraging and thoughtful review. We have answered your questions!**
>
> Thank you for your encouraging and thoughtful review. Below we address your comments:
>
> 1) Thanks for pointing to the infimal convolution view in convex analysis. This is a very interesting reformulation and we look forward to exploring its implications. About your question, we believe that while the objective function is important, the specific choice of the training algorithm has a great impact on the final performance.
>
> 2) We believe that SAM improves over VI methods in terms of final performance (and perhaps even efficiency). Even though we have not made a formal comparison in the paper, we have tried optimizing PAC-Bayesian bounds in the past and have not observed such significant benefits. In our experiments, instead of optimizing \rho directly, we treat it as a hyperparameter.
>
> Minor comments / typos: Thanks for pointing these out. We will fix them all in the final revision.

---

### Official Review · AnonReviewer1 · 2020-10-29
**Interesting work. It would be good to investigate its property.**

**Rating:** 6
**Confidence:** 2

**Review:**

- Overview

This paper proposed a learning algorithm using the idea of flatness.
Basically, the algorithm was constructed by a first-order Taylor approximation of flatness and using only the main term.
The developed algorithm uses the gradient of the loss with steepest direction when updating the parameters.
The proposed algorithm is simple and fast to work.
It is also easy to incorporate into existing networks and algorithms and has updated accuracy in many models.

- Comments.

This is an interesting study, as it is the first application of flatness to a practical algorithm.
It is also excellent in that it is easy to implement and immediately applicable. The performance is also nice.

The validity of this approximation is a concern.
When the authors perform a first-order Taylor approximation, they are looking only at the gradient of the parameter to determine the adversarial direction.
So, if the loss function is very spiky, there is a concern that the approximation will not work because the local gradient alone will not find the appropriate direction.
The nice networks used in their experiments is supposed to use smoothing techniques for the loss surface, such as the batch normalization and the skip connections, but the concern is whether the proposed method works properly in neural networks with a more less smooth loss surface.

To address this concern, my suggestion is to use the algorithm in without the Taylor approximation make comparisons on a more primitive network with a less smooth loss surface.
That way, users would feel more comfortable using the method.

---

> ### Author Response · Authors · 2020-11-17
> **Thanks for your review. We have addressed your concern.**
>
> Thank you for your review. We are glad that you liked our paper. Below, we address your only concern with the paper. If you find our response adequate, we would appreciate it if you increase your score.
>
> _The validity of this approximation is a concern. [...] So, if the loss function is very spiky, there is a concern that the approximation will not work because the local gradient alone will not find the appropriate direction_: We agree that there is no worst-case guarantee for this approximation for arbitrary networks.  However, we follow here in the footsteps of algorithms such as SGD in the context of training neural networks.  SGD, for example, arises from principled theoretical underpinnings, but the theory largely does not apply when using SGD to train neural networks.  Indeed, there is no known guarantee that SGD will be able to find a high-quality local minimum of a non-convex loss landscape.  Nonetheless, SGD is widely considered to be a valuable method due to its empirically observed effectiveness in finding neural network parameters of high quality.  Analogously, we motivate SAM from principled theoretical underpinnings, and while the practical application of SAM (Algorithm 1) involves approximations that we haven’t (yet) fully analyzed theoretically (a great direction for future work!), our empirical results show that these approximations yield an effective algorithm of substantial practical utility.
>
> Furthermore:
>  - As the reviewer noted, we achieve state of the art results with modern architectures; the approximation unequivocally works well for a range of settings.
>  - The current literature also suggests that the smoothness of the loss landscape is necessary to obtain good results with SGD (as the reviewer pointed out, this is a motivation behind recurrent connections). As a result, it seems that SAM will benefit from all the “tricks” researchers have developed to make SGD work in the first place.
>  - Our work can be connected to adversarial training, except that we perform adversarial attacks on the weight space instead of the input space. One common method for adversarial training is the fast gradient sign method (FGSM), which is the equivalent of SAM in the input space (although the projection is done on the l_inf sphere and not the l_2 sphere). This method is widely used due to its empirical efficacy, even if there is also no guarantee that the adversarial attack is the best possible.
>
>
> _The nice networks used in their experiments is supposed to use smoothing techniques for the loss surface, such as the batch normalization and the skip connections, but the concern is whether the proposed method works properly in neural networks with a more less smooth loss surface.  To address this concern, my suggestion is to use the algorithm in without the Taylor approximation make comparisons on a more primitive network with a less smooth loss surface. That way, users would feel more comfortable using the method_: We added a new appendix analyzing SAM when several steps of projected gradient ascent are used to compute epsilon (thereby computing the inner maximization with higher fidelity). The conclusion of this additional experiment is that i) for most of the training, one projected gradient step (i.e., the Taylor approximation) is enough to get a good approximation of the epsilon found without the Taylor approximation, ii) this approximation becomes weaker near convergence, where doing several iterations of projected gradient ascent yields a better epsilon, and iii) the test accuracy is not strongly affected by the number of project gradient iterations, thus the Taylor approximation is enough in practical settings.
>
> Additionally, the loss landscape presented on the right-hand side of Figure 1 is obtained for a CNN without skip connections. In this representation of the loss, it is possible to see that SAM actually managed to find a wider minimum in the absence of skip connections, and that the minimum found when not using SAM is very “spiky” and not smooth.  Similarly, the Hessian spectra presented in Section 4.2 were computed for a WideResNet40-10 trained on CIFAR-10 with and without SAM, without using batch normalization.  These spectra again confirm that SAM finds a lower-curvature minimum of the loss landscape.
>
> We additionally run further experiments for CNNs without batch normalization and without residual connections (models from Frankle and Carbin, https://arxiv.org/abs/1803.03635). We used the same hyperparameters as for the paper’s WideResnet-28x10 experiments on CIFAR and obtain the following results:
>
> Conv-2: **14.5**(SAM), 15.2 (SGD);
> Conv-4: **8.2** (SAM), 9.1(SGD);
> Conv-6: **6.8** (SAM), 7.8 (SGD);
>
> As seen in these results, SAM again performs well in the absence of batch normalization or residual connections.

---

### Official Review · AnonReviewer3 · 2020-11-03
**Interesting work with good results, concern is on selecting the right $\rho$**

**Rating:** 7
**Confidence:** 4

**Review:**

Motivated by the connection between the flatness of minima and its generalization ability, the authors propose Sharpness-aware Minimization (SAM), which explicitly minimizes both loss value and loss sharpness during training deep neural networks. They find SAM improves generalization for a range of image classification tasks and provide robustness to label noise as well. They also introduce a new notion of sharpness named m-sharpness.

Strength:
* The paper is overall well written with clear motivation.
* The experiments are comprehensive and the results show clear improvement over non-SAM approaches or previous SOTA.

Weakness:
* There is no clear definition of the “sharpness“ that the algorithm tries to optimize. Given many existing definitions of the sharpness (e.g., [1]), it is not clear how the proposed measurement connects or differs with previous works.

* My major concern is about the usage of hyperparameter $\rho$:

a) The introduction of the dataset and model dependent hyperparameter $\rho$ and the need of grid-search before training makes the algorithm more tricky to work and sensitive to other hyperparameters and scale of $w$, e.g., when weight decay is applied, the norm of $w$ usually shrinks during training, and the same radius $\rho$ could be too large for a small scaled $w$ at the end of training in comparison with the $w$ at the beginning. This discrepancy would become larger when the number of epochs training gets larger.

b) The details for how to obtain the optimal $rho$ is not quite clear, e.g., smaller $\rho$ in sec 3.3. An ablation study on the sensitivity of $\rho$ regarding different dataset, model and noisy level would be useful.

c) The wall-clock training time of the SAM method is not discussed. A comprehensive of the cost (including hyperparameter search for $rho$.) would be helpful to have for evaluating the complexity of the method.


* The message conveyed in section 4.1 is not quite clear. Does each accelerator perform independent $\epsilon$ estimation? Is $epsilon$ obtained on each accelerator synchronized after their estimation? Does it indicate the SAM training is done better in model-parallel in small batches rather than data-parallel with large batches?


Suggestions:
1) To avoid the scaling issue of $\rho$, one suggestion would be considering optimizing the sharpness metric on the normalized loss landscape as described in [2]. In Figure 1, the authors adopt [2] for comparing the landscape of minimas obtained by non-SAM and SAM, so it might be intuitive to optimize this normalized sharpness directly, in which $\rho$ can be fixed and random direction is sufficient?


2) The benefit of flatness to the robustness to label noise is not well discussed. What is the performance when the label noise is over 90% or even 100%. Eventually all models should not generalize given 100% corruption but it would be interesting to know where the limit of SAM is.

Minor:
* Some figures are not well described, e.g., the meaning of Figure 1 left is not quite clear.  Figure 2 is not intuitive as the loss contour value is not clear. It is not straightforward to know why w_{t+1}^{SAM} is a better or “flatter” move. The notion $w_{adv}$ is also not defined anywhere.

[1] Keskar et al, On large-batch training for deep learning: Generalization gap and sharp minima, ICLR 2017
[2] Li et al, Visualizing the Loss Landscape of Neural Nets, NIPS 2018

======
After Rebuttal

Thanks for the detailed reply and additional experiments. I increased my score accordingly and I hope the authors could further address following issues:

- While the results in C.3 shows default $\rho$ improves over SGD on most experiments (may also add SVHN and Fashion), I can still see its sensitivity to datasets, architecture, noise level and number of accelerators as shown in Table 6, 7, 8 and Fig. 3. For example, 0.05 is not close to optimal with labe noise 20%~60% in Table 8. It is unclear whether $\rho$ is robust to other hyperparameter changes (e.g., weight decay that controls weight scales).  So an ablation study on the sensitivity of $\rho$ and further explanation would be necessary and much valuable for practitioners.

- It would be also helpful if the authors can provide more details about how to get the flat minima of Fig .1 (right) when optimizing deep non-residual networks, such as $\rho$ and other hyperparameters.

- Minor: Table 8 should be validation errors rather than accuracy.

---

> ### Author Response · Authors · 2020-11-17
> **Thanks for your encouraging review. We have addressed all your comments about the tuning of the method (part 1)**
>
> Thanks for your encouraging and valuable feedback. Below, we address your comments:
>
> ### Definition of sharpness:
>
> In the paragraph after Theorem 1, we note that *The term in square brackets captures the sharpness of $L_S$ at $w$ by measuring how quickly the training loss can be increased by moving from $w$ to a nearby parameter value; this sharpness term [...]*, giving the definition of the sharpness that we use. The same definition of sharpness is used in the theorem: as we defined “sharpness” as $max_\{||\epsilon||_2\leq \rho} L(w+eps) - L(w)$, theorem 1 can read “$L_D <= L_s$ + sharpness + h(...). We have reworded the explanation of theorem 1 to make it more explicit that this is how we define sharpness.
>
> ### Shrinking of $w$
>
> While we agree that that could in principle occur, our empirical results (including several state of the art results on widely-studied benchmarks) unequivocally show that using a value of $\rho$ that is constant during training is sufficient to significantly increase the generalization of the learned model (all experiments hold $\rho$ constant throughout training).  Of course, modifying $\rho$ during training could potentially provide yet more gains; we believe that studying this possibility would be an excellent direction for future work.
>
> Regarding the scaling of the loss, the reviewer write: *To avoid the scaling issue of ρ, one suggestion would be considering optimizing the sharpness metric on the normalized loss landscape as described in [2]. In Figure 1, the authors adopt [2] for comparing the landscape of minimas obtained by non-SAM and SAM, so it might be intuitive to optimize this normalized sharpness directly, in which ρ can be fixed and random direction is sufficient?*: This is a great suggestion and we agree that there is probably some invariants that we could leverage in order to make the tuning of $\rho$ even easier. Because the optimal value of $\rho$ says something about the geometry of the loss landscape, we believe this is a very interesting direction for future research. We however don’t believe that random directions are promising; for example, Figure 6 in the appendix shows that random directions are inferior to adversarial directions in the setting considered.
>
>
> ### About the tuning of $\rho$
>
> Apologies for the lack of clarity. For the cifar experiments, we perform a standard hyper-parameter search on $\rho$ (from 3.1: _we tune via a grid search over {0.01,0.02,0.05,0.1,0.2,0.5} using  10%  of  the  training  set  as  a  validation  set_). All other hyper-parameters are held constant when tuning $\rho$ (such that we tune the learning rate and weight decay first, for the SGD baseline, then we only tune $\rho$ for SAM). For the Imagenet experiments, we searched for the optimal $\rho$ using the smallest model (Resnet-50) trained for only 100 epochs (from 3.1: _we use $\rho$= 0.05 determined via a grid search on ResNet-50 trained for 100 epochs_). We then used the same value of $\rho$ for all our imagenet experiments. For the fine tuning experiments, we also grid-searched $\rho$ for the Efficientnet-b7 model in the same fashion, using the validation set when available. We then used the best value of $\rho$ found for Efficientnet-l2, without additional tuning.  For the noisy label experiment, we also found $\rho$ by cross-validation. Feel free to check the scores on the validation set (table 8 of the revised paper) if that is of interest.
>
> SAM introduces only a single additional scalar-valued hyperparameter, $\rho$, and our results show that $\rho$ can be simply and effectively set via a standard grid search using a validation set.  This means that setting the value of SAM’s hyperparameter ($\rho$) is identical to the means by which hyperparameters are commonly set for a wide variety of other widely used and impactful procedures, such as dropout (which requires tuning dropout probability), MixUp (which requires tuning the mixing hyperparameter alpha), and even SGD (which requires tuning learning rate).  For instance, for the noisy label experiments, all of the methods to which we compare SAM similarly require their own hyperparameters to be tuned.
>
> However we understand the concerns of the reviewer and agree that a big sensitivity to the choice of hyper-parameters would make a method less easy to use. **To demonstrate that SAM performs even when $\rho$ is not finely tuned, we compiled the table for the cifar experiment, the imagenet experiment, and the fine tuning experiment using $\rho$=0.05 everywhere** and added those to our rebuttal. These results can be seen in appendix C.3 and show that **SAM is superior to SGD even when $\rho$ is set to a default value.**

---

> ### Author Response · Authors · 2020-11-17
> **Thanks for your encouraging review. We have addressed your additional concerns (part 2)**
>
> (follow up to https://openreview.net/forum?id=6Tm1mposlrM&noteId=RTA-IukmEl9)
>
> ### Wall clock and cost of tuning:
>
> **The wall-clock training time of the SAM method is not discussed.**: SAM does 2 forward-backward passes per step (vs. 1 for SGD) so SAM is at most 2x as slow. In practice, SAM does not need to load and preprocess a new batch or synchronize replicas between its 2 forward-backward passes, so SAM ends up being 1.5 to 2 times as slow as SGD (depending on the model, data pipeline, etc…). To control for that in Table 1, we also trained the SGD baseline for twice as long and reported the best results for SGD between n and 2n epochs. The ImageNet experiments report results for multiple different numbers of epochs, so that the reader can easily compare a model trained with SAM for n epochs and a model trained with SGD for 2n epochs (thereby enabling an “apples-to-apples” comparison).
>
> **A comprehensive of the cost (including hyperparameter search for rho) would be helpful to have for evaluating the complexity of the method.**  For the CIFAR/ImageNet experiment, we believe that we have reported enough elements to make this comparison possible, by reporting the best accuracy between n and 2n epochs for the baseline for CIFAR, and by reporting accuracies for 100, 200, 400 epochs for Imagenet. Regarding the hyper-parameters, SAM introduces a single one (which is in line with most deep learning research, including for instance MixUp (alpha), Group Normalization (number of groups), ShakeShake/ShakeDrop (several hyper-parameters), etc…) and we show in appendix C.3 that SAM yields non-trivial gains even if a single identical value of rho is used across all experiments.
>
>
> ### About the m-sharpness:
>
> **The message conveyed in section 4.1 is not quite clear. Does each accelerator perform independent ϵ estimation? Is epsilon obtained on each accelerator synchronized after their estimation?**  Each accelerator computes its own $\epsilon$ and the gradient at $w+\epsilon$ on its own sub-batch. Those per-accelerator gradients are then synced as usual before passing them to the optimizers.  In light of this comment, we are working on further clarifying Section 4.1 and intend to update the paper draft shortly.  We also hope that the codebase we are currently working on open sourcing will also provide additional clarity.
>
> **Does it indicate the SAM training is done better in model-parallel in small batches rather than data-parallel with large batches** : We believe these choices should be informed by the efficiency of the parallelism. In our experiments, we simply did not sync epsilon across accelerators (in the same way most codebases do not sync the batch norm statistics across accelerators for instance), as this reduces communication costs.  This approach additionally benefits generalization, as Section 4 explains.
>
> ### About the label noise experiments
>
> **The benefit of flatness to the robustness to label noise is not well discussed. What is the performance when the label noise is over 90% or even 100%. Eventually all models should not generalize given 100% corruption but it would be interesting to know where the limit of SAM is.**  While this paper focuses mostly on generalization (test error on classification tasks), we added the section about label noise to highlight additional benefits and interesting behavior of SAM. The noisy label setting of Section 3.3 is the same as that of Jiang et al (2019), which dictated the corruption ratios that we used. This experimental setting has been used in a variety of other published papers on learning with label noise.  That said, we certainly agree with the reviewer that additional experiments here could be of interest, and we would like to pursue them in future work.
>
>
> ### Conclusion
>
> We thank you again for your valuable feedback. We believe we have addressed all your concerns, especially with respect to the tuning of $\rho$ and in light of the results we added to appendix C4. As a result, we would really appreciate it if you consider increasing your score.

---

### Public Comment · ~Andrew_Gordon_Wilson1 · 2020-11-12
**Impressive empirical results but key citations and comparisons are omitted**

I am impressed by the empirical results in this paper and the relative simplicity of this approach. However, it would greatly strengthen the paper to have a proper discussion of and comparison with stochastic weight averaging (SWA) (https://arxiv.org/pdf/1803.05407.pdf, UAI 2018), which (1) also works by finding flat regions of the loss, (2) is less expensive and simpler than SAM, (3) in some cases leads to similar relative gains, (4) is widely used and has been published for a couple years, and (5) is trivially easy to compare against and is one of only a handful of optimizers natively supported by PyTorch. Loss surface flatness is a big part of the story behind SWA, and has been specifically exploited with SWA for improvements in (i) supervised learning; (ii) semi-supervised learning; (iii) reinforcement learning; (iv) Bayesian inference; (v) parallelized optimization, (vi) language modelling; (vii) generative modelling; (viii) low-precision training, and more. The PyTorch website summarizes many of these applications and papers: https://pytorch.org/blog/pytorch-1.6-now-includes-stochastic-weight-averaging/. It could additionally be useful to see what happens when we combine SAM with SWA.

---

> ### Author Response · Authors · 2020-11-17
> **Thanks for the reference. We have cited and compared with it.  SAM improves over SWA!**
>
> We thank you for your comment. We wholeheartedly agree that SWA, as a well-known and relevant algorithm, should be discussed in the paper, and we apologize for that unintended omission in the current version of the draft; we are updating the manuscript to make this addition. We further address your comments below:
>
> _SWA is less expensive and simpler than SAM_: In order to compare the costs of running SAM and SWA, we first note that SAM does 2 forward-backward passes per step (vs. 1 for SGD) so SAM is at most 2x as slow as SGD. However, because there is no IO cost or synchronization needed between replicas when computing eps and L(w+eps), we observed that in practice SAM turns out to often be faster than that, between 1.4x and 2x as slow as SGD depending on the model and data pipeline. In the SWA paper, the best results are obtained for 1.5x budget of compute, which is thus near what SAM requires.
>
> We have been additionally pleased to find that SAM is very simple to use.  Indeed, SAM can be utilized by dropping as few as 3 lines of code into one’s training loop, as seen in this modification of a JAX example found at https://github.com/google/flax/blob/master/examples/mnist/mnist_lib.py:
>
> ```
> @jax.jit
> def train_step(optimizer, batch):
>   """Train for a single step."""
>   def loss_fn(model):
>     logits = model(batch['image'])
>     loss = cross_entropy_loss(logits, batch['label'])
>     return loss, logits
>   grad_fn = jax.value_and_grad(loss_fn, has_aux=True)
>   (_, logits), grad = grad_fn(optimizer.target)
>
>   # ------------------ Add these lines for SAM ------------------ :
>   gnorm = jnp.sqrt(sum([jnp.sum(jnp.square(x)) for x in jax.tree_leaves(grad)]))
>   noisy_model = jax.tree_multimap(lambda a, b: a + RHO * b / gnorm, optimizer.target, grad)
>   (_, logits), grad = grad_fn(noisy_model)
>   # ------------------ End of SAM added code ------------------ .
>
>   optimizer = optimizer.apply_gradient(grad)
>   metrics = compute_metrics(logits, batch['label'])
>   return optimizer, metrics
> ```
>
>
> _SWA in some cases leads to similar relative gains_ : Comparing the models that are common to the SAM and SWA papers, we observe the following scores (relative gains in parentheses):
>
> ShakeShake** (1800) on cifar10, no augmentation:  SAM: 2.8* -> 2.3 (**18%**)    SWA: 3.07 -> 2.88 (6%)
> WRN-28-10 (200) on cifar10, no augmentations: SAM: 3.8* -> 2.7 (**29%**)    SWA: 3.82 -> 3.21 (16%)
> WRN-28-10 (200)  on cifar100, no augmentation: SAM: 18.8 -> 16.5 (**12%**)    SWA: 19.2 -> 17.8 (7%)
>
> (* our table report 3.5 (2.7) because we reported the score for 400 (3600) epochs for fair comparison with sam. See 3.1: _we allow each non-SAM training run to execute twice as many epochs as each SAM training run, and we report the best score achieved by each non-SAM training run across either the standard epoch count or the doubled epoch count_)
> (** SAM report results for the 2x96 model and SWA for the 2x64, thus the different accuracy for the baseline)
>
> We are definitely interested in running additional experiments using SWA for further comparison.
>
>
> _SWA is widely used and has been published for a couple years, and (5) is trivially easy to compare against and is one of only a handful of optimizers natively supported by PyTorch_ As we’ve discussed above, we would like to perform additional comparisons of SAM to even more algorithms (e.g., SWA, EntropySGD) in work going forward.

---

### Public Comment · ~Hikaru_Ibayashi1 · 2020-11-17
**Great application of sharpness but the proof of Theorem 2 seems to have a flaw**

As other people comment,  the algorithm is clean and the results of this work are undoubtedly great.
But I feel the proof of Theorem 2 seems to have a flaw.
This comment is partially similar to AnonReviewer2’s Minor Comments #10. But I suspect that there is a more critical issue.


In Appendix A.1, I assumed the authors claim that since (step 1) Eq. (13) holds with probability 1-δ and (step 2) ∥ε∥^2 < ρ with probability 1-δ, the desired bound holds with probability 1-2δ by union bound. (AnonReviewer2 comments that each substep should adopt δ/2 failure probability.) However, I suspect union bound cannot be applied here because these probabilities are on two different distributions. That means (step 1)’s failure probability is over the choice of the training set S ∼ D but (step 2)’s failure probability is over the randomness of ε.
(We can only obtain the following inequality at best Eεi∼N(0,σ)[LS(w+ε)]≤ (1-δ)max_{∥ε∥^2≤ρ} LS(w+ε) + δ. But this is not sufficient for the conclusion either.)


If I’m interpreting the proof in a wrong way, I would appreciate further clarification in the steps from (13) to the conclusion.


Minor typo
In the last equation of this proof, ≤ρ should be =: ρ^2 , otherwise, ρ would not be properly substituted to the inequality.

---

> ### Author Response · Authors · 2020-11-17
> **Thanks for pointing this out. This is fixed in the revision.**
>
> Thanks for reading our proof carefully. This was indeed a minor issue that was fixed in the revision in response to AnonReviewer2’s comment. Please see our response to AnonReviewer2 and updated paper for more details.

---

### Author Response · Authors · 2020-11-18
**Rebuttal Revision**

We thank the reviewers and the public comments for their encouraging comments and useful insights. We have made the following changes in the newly uploaded version (and will continue to improve the paper):

 - Following concerns regarding the sensitivity to $\rho$, we have run new experiments and recompiled the tables for the Cifar10/100, Imagenet, and finetuning experiments to report the accuracy obtained when using the default value of $\rho=0.05$, without additional tuning.  These results (in appendix C3) shows that **SAM outperform SGD even without tuning $\rho$**.
 - We finetuned our efficientnet on **Imagenet, leading to a new state of the art  (11.39% error rate for Efficientnet-l2-475**, compared to 11.8% without SAM).
 - We added additional experiments where we perform several steps of inner maximization (appendix C4). The main takeaway is that a single step of inner minimization is enough.
 - We fixed a minor flaw in the proof of the theorem. We thank the reviewers who looked at the proof carefully, even if it was in appendix.

---

### Comment · ~Juntang_Zhuang1 · 2021-05-03
**What's the difference from adversarial weight perturbation?**

Hi, congrats on the nice paper. However I'm confused by your paper and Adversarial Weight Perturbation [1]. It seems the algorithm is very close, except the perturbation is scaled by norm of gradient (Eq.2 in your paper) or norm of weight (Eq.10 in [1]), and the theorem and proof seems the same to me (both are the application of PAC-Bayes). Could you explain the difference? Did I miss anything? Thanks a lot.

[1] Wu de tal. Adversarial Weight Perturbation Helps Robust Generalization

---

> ### Comment · ~Pierre_Foret1 · 2021-05-05
> **Papers have a different focus**
>
> Hi! Thank your for your interest in our paper. We discussed "Adversarial Weight Perturbation Helps Robust Generalization" in the related work section of a new version of the paper (https://arxiv.org/pdf/2010.01412v2.pdf), unfortunately we forgot to add this discussion to the camera ready version. We will make sure to correct that soon. Wu et al is definitely a very interesting paper that we want to discuss.
> There are indeed several key differences between the papers:
> - Wu et al focus on adversarial robustness, while we focus on generalization in the "classical" non-adversarial setting. We aim to demonstrate that SAM could be added on top of modern, already well regularized networks, in order to strongly improve generalization on a large suit of datasets.
> - Whereas Wu et al consider a per-dataset perturbation (_depends on the entire samples (at least the batch samples) to make whole loss (not the loss on each sample) maximal_), our experiments on m-sharpness hint that a per example perturbation is optimal. While this difference can seem minor at first glance, we believe that it is very significant, as most generalization measures consider a per-dataset definition of sharpness. In practice, this manifests by the need to train on several replicas, with non-synced perturbations, to take full advantage of SAM.
> - Regarding the theorem and the proof, we agree that they are in essence the same in both papers. However we did not manage to find the theorem or the proof in the version of Wu et al available before our work was submitted to ICLR (https://arxiv.org/pdf/2004.05884v1.pdf), thus we did not discuss it in the paper.
> - As you noticed, there is also a difference in the scaling of the perturbation,  but further experiments would be required to analyze how significant this difference is.
>
> We hope this answer your question, please do not hesitate if you need any additional information.

---

### Comment · ~Martin_Jaggi1 · 2021-06-04
**known as extragradient method**

One could add that this method is simply called the **extragradient** method, since (Korpelevich 1976). A key aspect here however seems to be the use of a backwards instead of forward extrapolation stepsize. Several recent works study extragradient methods for deep learning, unfortunately not mentioned here, see e.g. the related work section in https://arxiv.org/abs/2006.05720 or maybe newer ones too.

---

### Decision · Program_Chairs · 2021-01-07
**Final Decision**

**Decision:**

Accept (Spotlight)

**Comment:**

The paper proposes to minimize the loss while regularizing its sharpness: so that the minimum will lie in a region with uniformly low loss.
The reviewers uniformly appreciated the paper. They have made a number of suggestion for improving the paper, which the authors should consider incorporating in their final version.